# Secreted dengue virus NS1 from infection is predominantly dimeric and in complex with high-density lipoprotein

Bing Liang Alvin Chew[1,2†], AN Qi Ngoh[3†], Wint Wint Phoo[4†], Kitti Wing Ki Chan[3†], Zheng Ser[4], Nikhil K Tulsian[5,6], Shiao See Lim[3], Mei Jie Grace Weng[1,2], Satoru Watanabe[3], Milly M Choy[3], Jenny Low[3,7], Eng Eong Ooi[3,8,9], Christiane Ruedl[10], Radoslaw M Sobota[4], Subhash G Vasudevan[3,11,12], Dahai Luo[1,2*]

[1]Lee Kong Chian School of Medicine, Nanyang Technological University, Singapore, Singapore; [2]NTU Institute of Structural Biology, Nanyang Technological University, Singapore, Singapore; [3]Program in Emerging Infectious Diseases, Duke-NUS Medical School, Singapore, Singapore; [4]Functional Proteomics Laboratory, Institute of Molecular and Cell Biology, Agency for Science, Technology and Research, Singapore, Singapore; [5]Department of Biological Sciences, National University of Singapore, Singapore, Singapore; [6]Singapore Centre for Life Sciences, Department of Biochemistry, National University of Singapore, Singapore, Singapore; [7]Department of Infectious Diseases, Singapore General Hospital, Singapore, Singapore; [8]Yong Loo Lin School of Medicine, National University of Singapore, Singapore, Singapore; [9]Saw Swee Hock School of Public Health, National University of Singapore, Singapore, Singapore; [10]School of Biological Sciences, Nanyang Technological University, Singapore, Singapore; [11]Department of Microbiology and Immunology, National University of Singapore, Singapore, Singapore; [12]Institute for Glycomics (G26), Griffith University Gold Coast Campus, Southport, Australia

*For correspondence:
luodahai@ntu.edu.sg

†These authors contributed equally to this work

Competing interest: The authors declare that no competing interests exist.

**Abstract** Severe dengue infections are characterized by endothelial dysfunction shown to be associated with the secreted nonstructural protein 1 (sNS1), making it an attractive vaccine antigen and biotherapeutic target. To uncover the biologically relevant structure of sNS1, we obtained infection-derived sNS1 (isNS1) from dengue virus (DENV)-infected Vero cells through immunoaffinity purification instead of recombinant sNS1 (rsNS1) overexpressed in insect or mammalian cell lines. We found that isNS1 appeared as an approximately 250 kDa complex of NS1 and ApoA1 and further determined the cryoEM structures of isNS1 and its complex with a monoclonal antibody/Fab. Indeed, we found that the major species of isNS1 is a complex of the NS1 dimer partially embedded in a high-density lipoprotein (HDL) particle. Crosslinking mass spectrometry studies confirmed that the isNS1 interacts with the major HDL component ApoA1 through interactions that map to the NS1 wing and hydrophobic domains. Furthermore, our studies demonstrated that the sNS1 in sera from DENV-infected mice and a human patient form a similar complex as isNS1. Our results report the molecular architecture of a biological form of sNS1, which may have implications for the molecular pathogenesis of dengue.

## eLife assessment

This potentially **useful** study aims to advance our understanding of the structure of the native form of a viral toxin secreted from infected cells. While some of the findings confirm previous reports, the new claims in this study are unfortunately only **inadequately** supported by the methods and

analyses used. More rigorous approaches are needed to justify the main conclusion that the structure of the viral toxin derived from infected cells in this study is distinct from previously reported structures of recombinantly expressed versions of the toxin.

## Introduction

Dengue virus (DENV) is a member of the flavivirus genus and is the cause of significant healthcare problems and economic burden worldwide, partially due to the lack of effective therapeutics and the limited efficacy of licensed vaccines, Dengvaxia and QDENGA (*Bhatt et al., 2013*; *Diamond and Pierson, 2015*). While the majority of DENV infections are mild or asymptomatic, severe dengue infections, marked by vascular leakage, can be life-threatening and even fatal. The severity of dengue infections has been demonstrated to be correlated to high sera levels of sNS1 in clinical studies (*Libraty et al., 2002*; *Paranavitane et al., 2014*). The viral nonstructural protein 1 (NS1) is a highly conserved and multifunctional protein that exists as intracellular membrane-associated NS1, a presumed GPI-anchored outer plasma membrane-associated NS1, and secreted NS1 (sNS1) during viral infection (*Glasner et al., 2018*). NS1 could interact with a myriad of proteins associated with its immune evasion and virotoxin roles (*Watterson et al., 2016*). The interactions of NS1 with viral E protein (*Scaturro et al., 2015*) and the NS4A-2K-NS4B precursor protein *Płaszczyca et al., 2019* have been shown to be important in virus production and replication, respectively. Intracellular NS1 found in the endoplasmic reticulum (ER) lumen is an essential part of the membranous viral RNA replication compartment (*Scaturro et al., 2015*). Upon release from the cells, sNS1 enters the blood circulation where in vitro and in vivo mouse evidence demonstrated its ability to induce vascular permeability, a hallmark of severe dengue, either independently (*Puerta-Guardo et al., 2016*) or by inducing pro-inflammatory responses (*Beatty et al., 2015*).

The high-resolution crystal structures of NS1 dimer have been reported for DENV (*Akey et al., 2014*), West Nile Virus (WNV) (*Akey et al., 2014*), and Zika Virus (ZIKV) (*Xu et al., 2016*; *Brown et al., 2016*), which provide a conserved molecular view of flaviviral NS1. Mature NS1 is 352 amino acids long with an apparent molecular mass between 40 and 50 kDa depending on its glycosylation state. NS1 has a three-domain architecture, a hydrophobic β-roll (residues 1–29), an α/β wing (38–151), and a β-ladder (181–352). The connector segments between the wing and β-ladder domains, residues 30–37 and 152–180, form a three-stranded β-sheet. The dimer has a distinct cross shape with the wings extending from the central β-ladder, which has an extended β-sheet that faces the hydrophobic β-roll and a 'spaghetti loop' on the opposite hydrophilic outer face that lacks structured elements (*Akey et al., 2014*; *Xu et al., 2016*; *Brown et al., 2016*). sNS1 was reported to be a barrel-shaped hexamer with lipid cargo held together by hydrophobic interactions based on biophysical and low-resolution EM analysis (*Flamand et al., 1999*; *Gutsche et al., 2011*; *Muller et al., 2012*). Recent cryoEM structures of recombinant sNS1 (rsNS1) from DENV-2 showed mainly tetrameric rsNS1, with only 3% of the population found to be in the hexameric form *Shu et al., 2022*. While earlier studies by Gutsche et al, (2011) found that the lipid profile of DENV-1 infection-derived sNS1 to be similar to high-density lipoprotein (HDL) *Gutsche et al., 2011*, the same research group now showed that DENV-2 rsNS1 could dock onto HDL with direct visualization using negative stain EM analysis as reported by *Benfrid et al., 2022*. Additionally, sNS1 appears to utilize the scavenger receptor B1, a known receptor for HDL, as the cognate receptor in cultured cells (*Alcalá et al., 2022*). Since the interactions of sNS1 with the target cells implicated in pathogenesis require the hydrophobic β-roll side to be exposed (*Akey et al., 2014*), the mechanism by which the oligomeric sNS1 with its lipid load in the central channel dissociates and associates has been a topic of intense research interest (*Benfrid et al., 2022*; *Avirutnan et al., 2007*; *Alcalá et al., 2017*). However, the native structure of the extracellular sNS1 during viral infection remains unclear.

To closely mimic sNS1 circulating in dengue patients' sera, we obtained infection-derived sNS1 (isNS1) from the culture supernatant of Vero cells infected with either the DENV2 WT (isNS1wt) or T164S mutant (isNS1ts). The T164S mutation in the greasy finger loop between the wing and β-ladder interdomain of NS1 was identified from a DENV2 epidemic in Cuba in 1997, where a correlation with enhanced clinical disease severity was observed (*Rodriguez-Roche et al., 2005*). We demonstrated that this single mutation in NS1 could directly cause lethality in mice and increased sNS1 secretion

(*Chan et al., 2019*), which served as an epidemiologically relevant tool to study the biochemical and structural characteristics of sNS1 in this study.

We determined that the isNS1 is a complex of the NS1 dimer embedded on a single HDL particle composed of ApoA1. By applying integrative structural tools, we report the cryoEM structural model of the purified isNS1:HDL complex at ~8 Å resolution with protein–protein interactions between NS1 and ApoA1 defined by crosslinking mass spectrometry (XL-MS). Furthermore, the Ab56.2 binding site model in the cryoEM model of isNS1:ApoA1 complex was elucidated using hydrogen deuterium exchange mass spectrometry (HDX-MS) as well as NS1 peptide competition ELISA. Interrogation of DENV-infected mouse sera and a human patient suggests that the sNS1:HDL complex exists in vivo as a similar complex to isNS1.

## Results
### Native isNS1 is a complex of NS1 with ApoA1

We purified isNS1 using a monoclonal antibody 56.2 from the supernatant of the infected Vero cell cultures (*Figure 1a*, *Figure 1—figure supplement 1a and b*). Initial negative stain screening of the isNS1wt (*Figure 1—figure supplement 1c*) showed some 2D classes of NS1 dimers presumably docked onto a spherical density, mirroring the in vitro reconstitution of rsNS1 with HDL complex negative stain microscopy results by *Benfrid et al., 2022*. Both immunoaffinity-purified isNS1wt and isNS1ts retained a molecular size of ~250 kDa as detected with Coomassie blue and on a western blot using Ab56.2 following separation on a Native-PAGE (*Figure 1b*, *Figure 1—source data 1 and 2*, *Figure 1—figure supplements 2 and 3*, *Figure 1—figure supplement 2—source data 1–3*), as previously reported (*Chan et al., 2019*). The isNS1wt migrated slower than the rsNS1 protein (a gift from *Shu et al., 2022*), indicating possible conformational or oligomeric differences between the two NS1 species (*Figure 1b*). On a reducing SDS-PAGE, we identified two major bands of approximately 50 kDa and 25 kDa in isNS1wt, the former corresponding to the single monomeric NS1 band seen in rsNS1 (*Figure 1c*, *Figure 1—source data 3*). Preliminary mass ID identified the 25 kDa protein as bovine apolipoprotein A1 (ApoA1), a major component of HDL in bovine serum-containing culture media, which has a reported molecular weight of 28 kDa (*Kitti, 2019*). Hence, the two bands were further validated on a western blot, which confirmed the 50 kDa band as NS1 and the 25 kDa band as ApoA1 respectively (*Figure 1c*, *Figure 1—source data 4 and 5*). The same observations were seen in isNS1ts (*Figure 1—figure supplement 3b*, *Figure 1—source data 3–5*).

Next, we elucidated the composition of the prominent 250 kDa band observed in the Native-PAGE by label-free quantification (LFQ) analysis of proteins using liquid chromatography-mass spectrometry (LC-MS). We obtained LC-MS data for the ~250 kDa protein excised from the Native-PAGE separation of the immunoaffinity-purified isNS1wt (represented in *Figure 1b*, purple box as Gel Band 1); an aliquot of the elute fraction from immunoaffinity purification (depicted in *Figure 1a* blue box and referred to as 'elute') and the 50 kDa and 25 kDa bands excised after separation of the elute fraction on a reducing SDS-PAGE (*Figure 1c*, represented by red and green boxes labeled as Gel Band 2 and 3, respectively) as positive controls. The raw data for the ion intensities of the samples tested are as detailed in *Supplementary file 1*. We found that the combined LFQ intensity of NS1 and ApoA1 accounted for more than 77.6 and 88.6% of the total ion intensity count (TIC) detected from the elute fraction (*Figure 1a*, blue box) and the 250 kDa band (*Figure 1b*, the purple box labeled '1'), respectively, indicating that NS1 and ApoA1 were the major proteins in the elute fraction and the ~250 kDa band from immunoaffinity-purified isNS1wt (*Figure 1d*). Accordingly, as depicted in *Figure 1d*, NS1 accounted for 96.85% of the TIC for the 50 kDa band (*Figure 1c*, red box '2'), while the intensity of ApoA1 accounted for 96.23% of the TIC for the 25 kDa band (*Figure 1c*, green box '3'). Similar LFQ analysis of LC-MS data for the purified isNS1ts mutant showed that NS1 and ApoA1 were the major components (*Figure 1—figure supplement 2*) of the respective elute and ~250 kDa gel band on Native-PAGE (*Figure 1—figure supplements 2a and 3a*, purple box). The corresponding 50 kDa and 25 kDa gel bands on the reducing SDS-PAGE (*Figure 1—figure supplements 2b and 3b*) identified as NS1 and ApoA1 by immunoblotting with the respective antibodies (*Figure 1—figure supplement 3b*) were further confirmed by LC-MS as NS1 and ApoA1 (*Figure 1—figure supplements 2c and 3c*, *Supplementary file 1*). Taken together, the results from the Native-PAGE and LFQ by LC-MS strongly

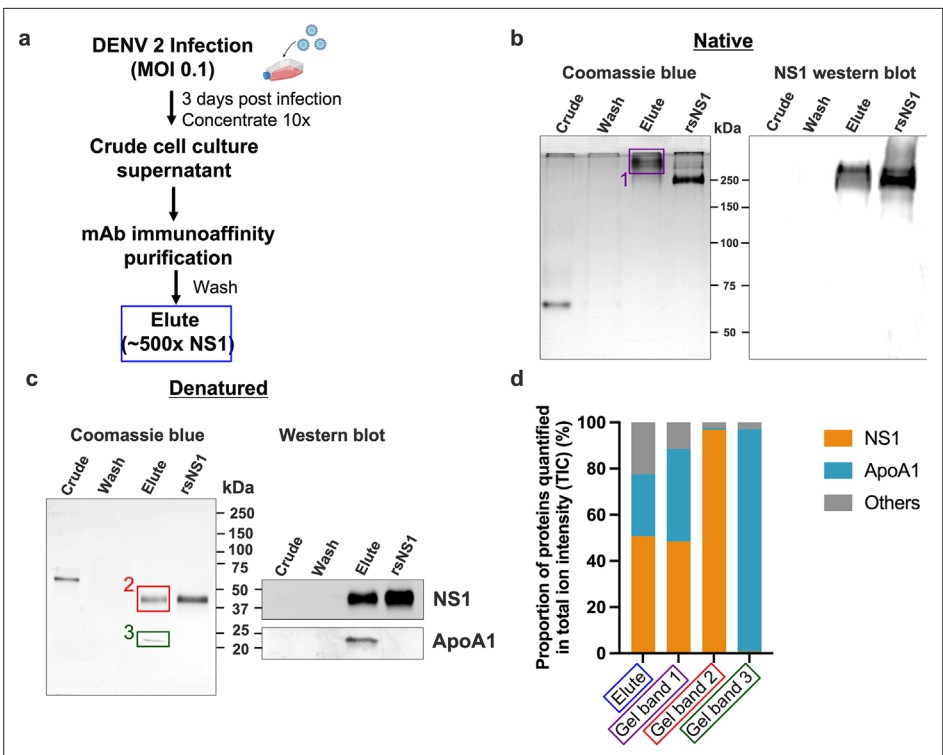

**Figure 1.** Composition of the secreted NS1 from dengue virus (DENV)-infected Vero cells. DENV 2 WT cell culture supernatant was filtered, supplemented with protease inhibitor cocktail and 0.05% sodium azide, concentrated using a 100 kDa MWCO Vivaflow cassette and purified using 56.2 anti-NS1 antibody immunoaffinity chromatography. The eluted isNS1wt was dialyzed against PBS, concentrated, and stored at –80°C until further use. (**a**) Schematic of isNS1 purification to illustrate the samples used for gel analyses. % NS1 is measured by the total amount of NS1 (quantified using the anti-NS1 ELISA kit [Bio-Rad] as a percentage of total protein [quantified using the Bradford assay]) found in each sample. Details of the % enrichment in NS1 along the purification process are as shown in *Figure 1—figure supplement 1b*. (**b**) Coomassie blue detection of proteins from crude, wash, and elute immunoaffinity fractions for isNS1wt, with the recombinant sNS1 (rsNS1) obtained from *Shu et al., 2022* as a positive control, after separation on a 10% Native-PAGE gel (left). The crude and elute fractions contain 1 μg of total protein. The wash fraction contains approximately 100 ng of total protein in maximum well volume of the gel. The same set of samples were also subjected to a western blot detection of NS1 using 56.2 anti-NS1 antibody after separation on a 10% Native-PAGE (right). The crude and elute fractions contain 500 ng of total protein. The wash fraction contains approximately 100 ng of total protein in maximum well volume of the gel. (**c**) Coomassie blue detection of proteins from crude, wash, and elute immunoaffinity fractions for isNS1wt and rsNS1 (*Shu et al., 2022*), after separation on a 4–20% reducing SDS-PAGE gel. The crude and elute fractions contain 1 μg of total protein. The wash fraction contains approximately 100 ng of total protein in maximum well volume of the gel. Similarly, the same set of samples were also subjected to a western blot detection of NS1 and ApoA1 using 56.2 anti-NS1 antibody or ApoA1 antibody (Biorbyt, orb10643), respectively, after separation on a 4–20% reducing SDS-PAGE (right). (**d**) In-gel protein identification of the purified isNS1wt by liquid chromatography mass spectrometry (LC-MS). Proportion of NS1, ApoA1 and other unidentified proteins quantified in total ion intensity, obtained from the following samples: elute in solution (boxed in blue), 250 kDa gel band (boxed in purple), 50 kDa gel band (boxed in red), and 25 kDa gel band (green). The boxed gel bands are from representative gels showing the different protein species found while the actual gel bands used for protein identification by LC-MS are as shown in *Figure 1—figure supplement 2a and b*.

The online version of this article includes the following source data and figure supplement(s) for figure 1:

**Source data 1.** Raw and annotated image for the PAGE gel stained in Coomassie blue.

**Source data 2.** Raw and annotated image for the western blot analysis (anti-NS1).

**Source data 3.** Raw and annotated image for the PAGE gel stained in Coomassie blue in *Figure 1c*.

**Source data 4.** Raw and annotated image for the western blot analysis (anti-NS1) in *Figure 1c*.

**Source data 5.** Raw and annotated image for the western blot analysis (anti-ApoA1) in *Figure 1c*.

*Figure 1 continued on next page*

*Figure 1 continued*

**Figure supplement 1.** Purification and negative stain electron microscopy screening of in vitro infection-derived sNS1 from infected Vero cells.

**Figure supplement 2.** Protein identification analysis of excised gel bands via liquid chromatography mass spectrometry (LC-MS).

**Figure supplement 2—source data 1.** Raw and annotated image for the PAGE gel stained in Coomassie blue for isNS1wt.

**Figure supplement 2—source data 2.** Raw and annotated image for the PAGE gel stained in Coomassie blue for isNS1ts.

**Figure supplement 2—source data 3.** Raw and annotated image for the SDS-PAGE gel stained in Coomassie blue for isNS1wt and isNS1ts presented in *Figure 1—figure supplement 2b*.

**Figure supplement 3.** Composition of the secreted isNS1ts from dengue virus (DENV)-infected Vero cells.

**Figure supplement 4.** CryoEM analysis for isNS1ts.

indicated that the isNS1 purified from the supernatant of infected cells in vitro is an approximately 250 kDa complex of NS1 and ApoA1.

## CryoEM structure of isNS1

To gain insights into the molecular organization of this native isNS1, we attempted cryoEM analysis of various isNS1 complexes (*Table 1*). The 2D classes of isNS1ts are observed to be heterogeneous spheres with some internal features, some of which resemble the cross-shaped NS1 protruding from the sphere (*Figure 1—figure supplement 4*). However, the 3D reconstruction of isNS1ts alone was observed to have no discernible features beyond a spherical-like density with varying dimensions averaging around 106 Å by 77 Å (*Figure 1—figure supplement 4b and c*). To obtain higher resolution structural information on the complex, we collected data for the ternary complexes of the isNS1wt:Ab56.2 isNS1wt:Fab56.2 (*Figure 2a*, *Figure 2—figure supplements 1 and 2*). The use of antibodies as fiduciary markers is a well-proven approach for solving the structures of small proteins (*Wu et al., 2012*; *Bloch et al., 2020*). Overall, both samples resulted in similarly distinguishable 2D class averages (*Figure 2b and c*), which show asymmetrical binding of one unit of Fab56.2 to the dimeric cross-shaped sNS1 associated with a lower-density sphere, presumably HDL.

In the isNS1wt:Fab56.2 ternary complex dataset, we further observed two apparent but rare subclasses of free sNS1wt dimer bound to two units of Fab56.2 representing 3.5% of the total population

**Table 1.** CryoEM data collection statistics.

|  | isNS1ts | isNS1wt:Ab56.2 | isNS1wt:Fab56.2 | isNS1ts:Fab56.2 |
|---|---|---|---|---|
| Grid type | gAU | gAU | gAU | gAU |
| Microscope | | Titan Krios G3 | | |
| Voltage (keV) | | 300 | | |
| Camera | Gatan K2 | Gatan K2 | Gatan K2 | Gatan K3 |
| Magnification (nominal) | 165,000 | 165,000 | 165,000 | 105,000 |
| Pixel size (Å/pixel) | 0.85 | 0.85 | 0.85 | 0.858 |
| Total electron dose (e-/Å$^2$) | 53 | 53 | 70 | 60 |
| Exposure rate (e-/Å$^2$/s) | 8.8 | 8.8 | 7 | 10.2 |
| Number of frames | 40 | 40 | 50 | 50 |
| Defocus range (μm) | 1.2–2.0 | 1.2–2.0 | 1.0–1.8 | 0.7–1.5 |
| Automation software | | EPU | | SerialEM |
| Energy filter slit width | | 20 eV | | |
| Micrographs collected (no.) | 3146 | 2556 | 2559 | 5443 |
| Micrographs post-clean (no.) | 3000 | 2072 | 2090 | 4241 |

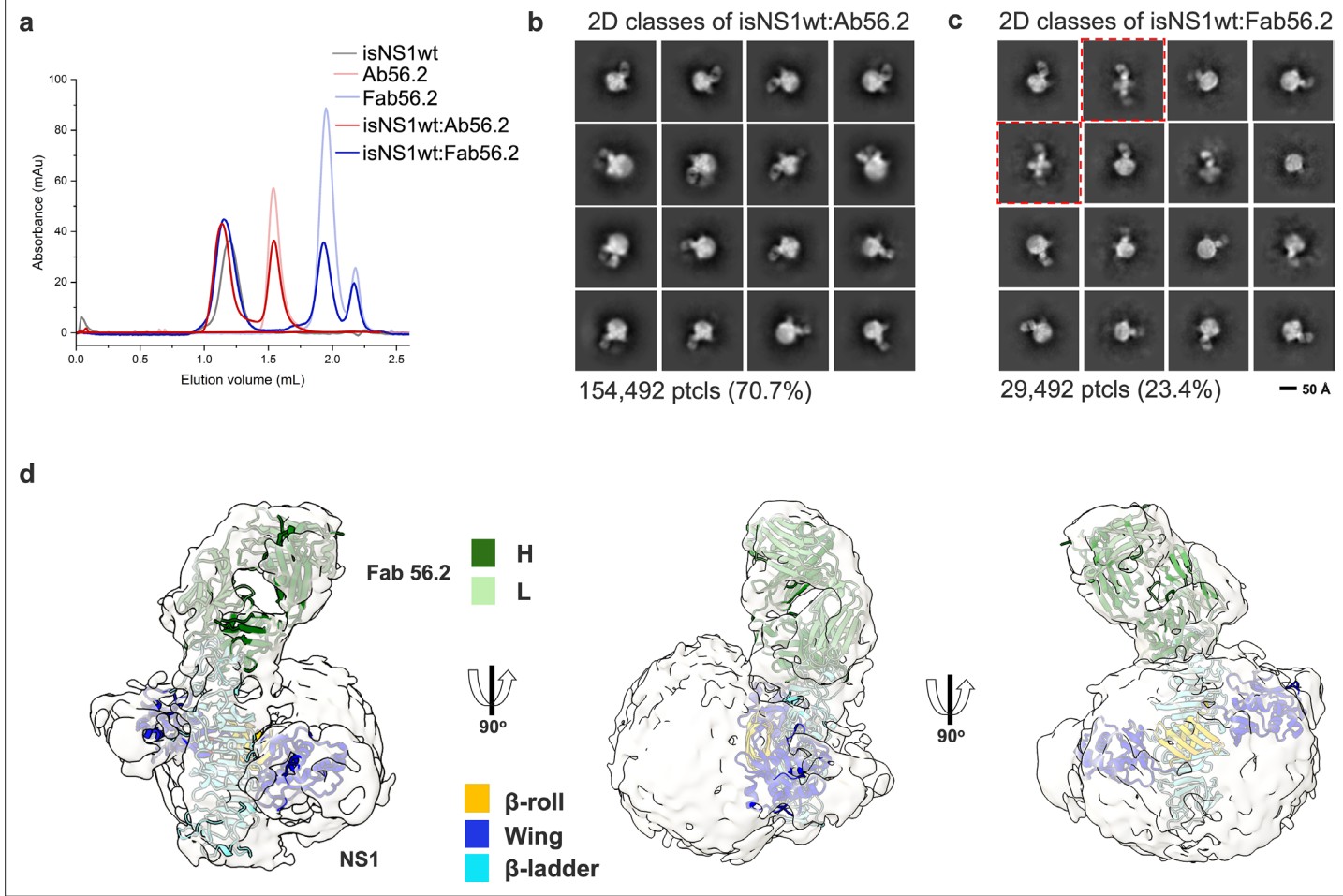

**Figure 2.** CryoEM analysis of secreted NS1 in complex with antibody Ab56.2. (**a**) Size-exclusion chromatography was run on a Superdex 200 increase 3.2/300 GL column connected to the ÄKTA purifier with a flow rate of 0.075 mL/min in PBS (pH 7.4) for purified isNS1wt (gray), Ab56.2 (faded red), Fab56.2 (faded blue), isNS1wt:Ab56.2 (red), and isNS1wt:Fab56.2 (blue). A slight leftward shift in elution volume was observed for isNS1wt upon complexing with Ab56.2 and Fab56.2. 2D class averages of (**b**) isNS1wt:Ab56.2 and (**c**) isNS1wt:Fab56.2 showing representative sub-class of the Fab56.2:isNS1:HDL particles with black scale bar, 50 Å. The corresponding number of particles and percentages are listed below the respective boxes. Red dashed line boxes highlight two rare views consisting of 1033 particles (3.5%) only seen in isNS1wt:Fab56.2 sample. (**d**) Model of isNS1wt:Fab56.2 predicted structures rigid body fitted in the cryoEM map of Fab56.2:isNS1:HDL (gray, contoured at 0.14) with correlation value of 0.75 to the fitted regions (map simulated from atoms at 5 Å). isNS1wt is colored by its three domains, namely the β-roll (orange), wing (blue), and β-ladder (cyan). Fab56.2 is colored by its heavy chain (dark green) and light chain (light green).

The online version of this article includes the following figure supplement(s) for figure 2:

**Figure supplement 1.** CryoEM analysis for isNS1wt in complex with Ab56.2.

**Figure supplement 2.** CryoEM analysis for infection-derived isNS1wt:Fab56.2.

of picked particles (*Figure 2c*, red dashed boxes). While the cryoEM map reconstructions remain limited at low resolutions (*Figure 2—figure supplements 1 and 2*), they could be fitted with the NS1 dimer and Fab56.2 models (*Figure 2d*) predicted separately using AlphaFold2 (*Jumper et al., 2021*; *Mirdita et al., 2022*), with a correlation value of 0.75 to the fitted regions using a simulated 5 Å map from the NS1 dimer and Fab models. The AlphaFold2 predicted model of the NS1 dimer was used despite the availability of the crystallographic dimer model (PBS ID: 6WER) (*Biering et al., 2021*) given their high similarity (root mean square deviation (RMSD): 0.67) and that the latter had a few unmodeled residues (aa 118–128). The map-model fitting showed that the epitope sites were at the tip of the C-terminal β-ladder region. The approximate dimensions of the spherical HDL could also be measured at 82 Å by 65 Å (*Figure 2—figure supplements 1 and 2*), which aligned with existing reports of spherical HDL being measured at 70–120 Å in diameter with predominantly homodimeric

ApoA1 arranged as an anti-parallel double-belt stabilized by intermolecular salt bridges (*Silva et al., 2008*; *He et al., 2019*). We predicted a model of an antiparallel dimer of ApoA1 using AlphaFold2 (*Jumper et al., 2021*; *Mirdita et al., 2022*) without the first 58 residues of ApoA1 as the N-terminal domain is highly flexible (*Silva et al., 2008*). The model has a cross-sectional distance of approximately 78 Å and fits into the spherical density in the cryoEM map. The NS1 dimer appears semi-embedded on the HDL particle through its hydrophobic surface, with a conformation that exposed only one end of the β-ladder domain to which Ab56.2 and Fab56.2 bind. Overall, the cryoEM model indicated that the purified ~250 kDa isNS1 is a complex of an NS1 dimer and HDL, consistent with the gel and LFQ analysis of LC-MS data presented above.

Similarly, we determined the structures of the ternary complex of isNS1ts mutant with Fab56.2 (*Figure 3a*). Interestingly, 2D class averages of isNS1ts:Fab56.2 dataset showed a significant sub-population of a free NS1ts dimer bound to two units of Fab56.2 (53.7%), in addition to the HDL spheres (23.6%) and isNS1ts dimer:HDL:Fab56.2 (22.7%) classes (*Figure 3b*, *Figure 3—figure supplement 1*). The predicted structure of the isNS1ts dimer bound to Fab56.2 could be fitted with an overall correlation value of 0.5 using map simulated from atoms at 5 Å (*Figure 3c*). Achieving atomic resolution to map the Fab binding interface with the NS1 β-ladder domain remains a challenge due to the preferred orientation (*Figure 3—figure supplement 1*). A comparison of the cryoEM density maps of free isNS1ts dimer:Fab56.2 (*Figure 3D*, gray) and isNS1ts dimer:HDL:Fab56.2 ternary complex (*Figure 3d*, yellow) fitted with a correlation value of 0.71 based on the NS1 map region revealed a pose difference of the Fab56.2 between the free isNS1ts dimer and the HDL-bound isNS1ts dimer (*Figure 3d*, inset). Further comparison of the density maps of the isNS1 dimer:HDL population in the Fab56.2 ternary complexes of isNS1wt (*Figure 3e*, purple) and isNS1ts (*Figure 3e*, yellow) revealed conformational similarities of the isNS1 dimer:HDL complex.

## XL-MS maps the inter- and intramolecular architecture of the isNS1

Next to refine our understanding of the NS1 dimer:HDL complex of the isNS1 indicated by the LFQ of LC-MS (*Figure 1d*, *Figure 1—figure supplement 3d*) and cryoEM results (*Figures 2d and 3d*), we probed the interactions between NS1 and ApoA1 – the major protein moiety of HDL – using XL-MS (*Kao et al., 2011*) on the immunoaffinity purification eluate fraction (*Figure 4a*, *Figure 4—source data 1*, *Figure 4—figure supplement 1a and b*, *Figure 4—figure supplement 1—source data 1 and 2*). We identified 28 NS1:NS1 and 29 ApoA1:ApoA1 intramolecular crosslinks, as well as 25 NS1:ApoA1 intermolecular crosslinks for isNS1wt (*Figure 4b*; *Supplementary file 2 and 3*). As depicted in *Figure 4b and c*, the NS1 residues that are involved in both intra- and intermolecular crosslinks were located mainly on the β-roll and wing domains, with ~34% of the NS1:ApoA1 crosslinks being located on the β-roll domain. The NS1:NS1 intramolecular crosslink pairs were further validated on the predicted NS1 dimer model by implementing a 30 Å cutoff distance for the Cα atoms of crosslinked residues, a criterion that was met by 85% of the NS1:NS1 intramolecular crosslinks (24 out of 28; *Figure 4d*). The NS1:ApoA1 intermolecular crosslinks for the immunoaffinity-purified isNS1wt were found to be located between the β-roll and wing domains of NS1 and the helices 3, 4, 8, and 10 of ApoA1 (*Figure 4b and d*). Applying the same cutoff value of <30 Å distance between the Cα atoms of the NS1:ApoA1 intermolecular crosslinked residues and validating it on the cryoEM model of isNS1wt in this study (*Figure 4d and e*) satisfied 76% of the crosslinks (19 out of 25 intermolecular crosslinks between NS1 dimer and ApoA1). In the case of ApoA1:ApoA1 intramolecular crosslinks, the <30 Å distance between their Cα atoms cutoff satisfied 62% of the crosslinks (18 out of 29 ApoA1:ApoA1 intramolecular crosslinks). The higher number of violated crosslinks in ApoA1:ApoA1 compared to those in NS1:NS1 could be due to the flexible and dynamic nature of ApoA1 dimers, which may prevent it from assuming a fixed relative position among the ApoA1 dimer population in the HDL. A similar crosslink pattern to isNS1wt is observed in the XL-MS mapping of isNS1ts mutant (*Figure 4—figure supplement 1c and d*) followed by rigid-body fitting into the corresponding cryoEM map (*Figure 4—figure supplement 1e*), except that there were fewer crosslink sites (*Figure 4—figure supplement 1f*, *Supplementary file 2 and 3*). This indicated that isNS1ts dimer may have a weaker affinity to HDL in comparison to the isNS1wt dimer. This finding is consistent with the cryoEM data for isNS1ts where we found that free isNS1ts dimers bound to Fab56.2 formed a significant sub-population (34.5%) of the total picked particles (*Figure 3b*, *Figure 3—figure supplement 1b*). Overall, the XL-MS data and structural validation agreed with the cryoEM model of

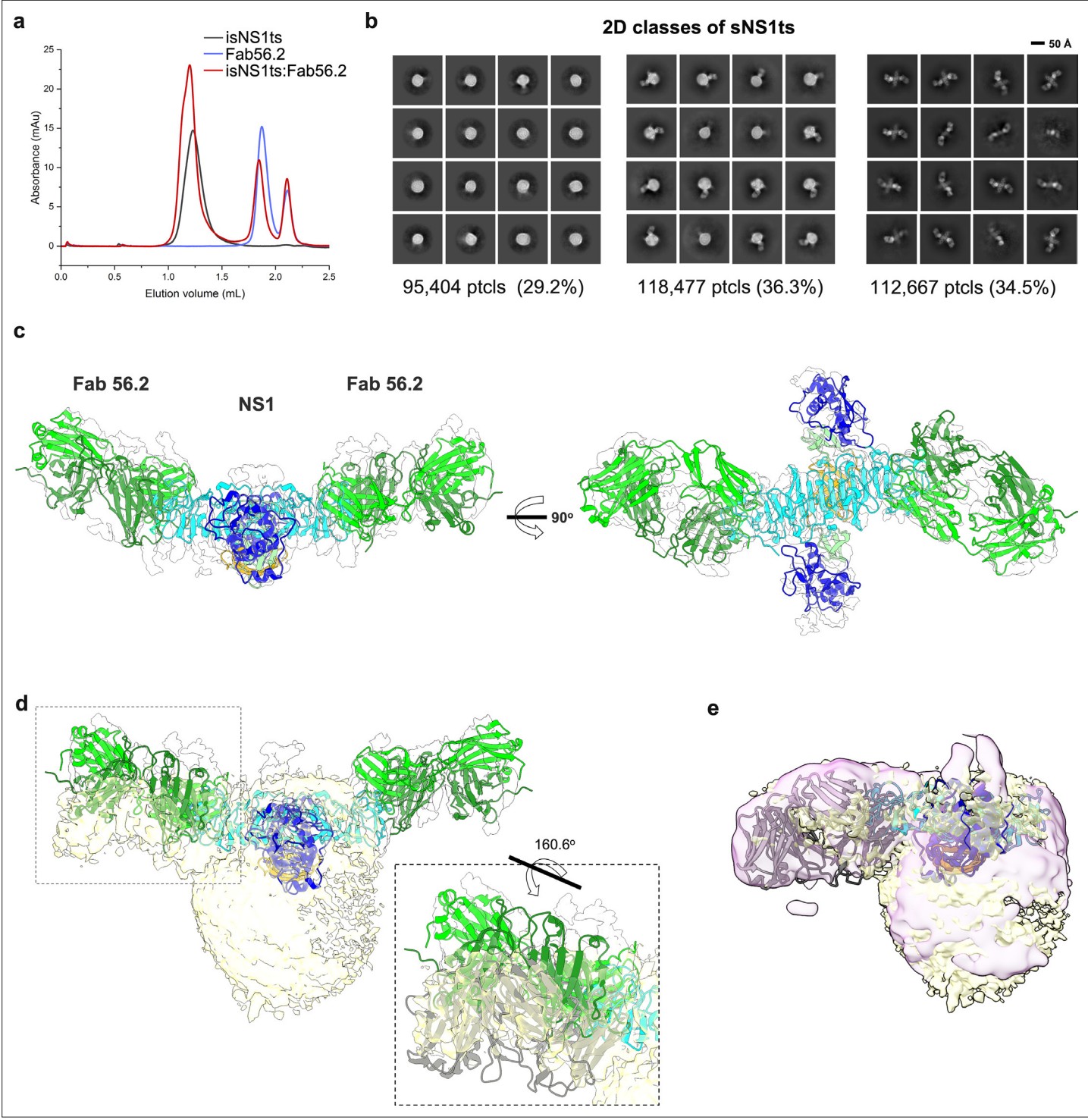

**Figure 3.** Secreted NS1 forms free dimers in complex with antibody Ab56.2. (**a**) Size-exclusion chromatography was run on a Superdex 200 increase 3.2/300 GL column connected to the AKTA purifier with a flow rate of 0.075 mL/min in PBS (pH 7.4) for purified isNS1ts (gray), Fab56.2 (faded blue), and isNS1ts:Fab56.2 (red). A similar leftward shift in elution volume was also observed for isNS1ts upon complexing with Fab56.2. (**b**) 2D class averages of isNS1ts:Fab56.2 dataset with 326,548 particles picked and separated into three distinct populations, high-density lipoprotein (HDL) spheres, Fab56.2:isNS1:HDL, and free isNS1:Fab56.2. Black scale bar, 50 Å, as indicated. The corresponding number of particles and percentages are listed below the respective boxes. (**c**) Model of isNS1ts dimer and Fab56.2 predicted structures rigid body fitted in the isNS1ts:Fab56.2 density map (gray, contoured at 0.14) with correlation value of 0.53 (overall, map simulated from atoms at 5 Å). isNS1ts is colored by its three domains, namely the β-roll (orange), wing (blue), and β-ladder (cyan). Fab56.2 is colored by its heavy chain (dark green) and light chain (light green). (**d**) Density map fitting between

*Figure 3 continued on next page*

*Figure 3 continued*

isNS1ts:Fab56.2 (gray, contoured at 0.14) to Fab56.2:NS1ts:HDL (yellow, contoured at 0.1) with correlation value of 0.53 (overall) and 0.7137 (on D2NS1 map region only). Inset shows the difference in the pose of Fab from the free NS1 form to the HDL-bound form. (**e**) Density map fitting between Fab56.2:isNS1wt:HDL (purple, contoured at 0.05) to Fab56.2:NS1ts:HDL (yellow) with correlation value of 0.72 (overall).

The online version of this article includes the following source data and figure supplement(s) for figure 3:

**Source data 1.** The corresponding PDB models for isNS1ts:Fab56.2 (EMD-36483) and Fab56.2:isNS1ts:HDL (EMD-36480).

**Figure supplement 1.** CryoEM analysis for isNS1ts in complex with Fab56.2.

isNS1 being a complex of the NS1 dimer embedded on a HDL particle composed of an ApoA1 dimer (*Figure 4d and e*).

Given the reported structures of hexameric and tetrameric rsNS1 from mammalian expression systems, insect cells (*Akey et al., 2014*; *Gutsche et al., 2011*; *Muller et al., 2012*; *Biering et al., 2021*), and Expi293 HEK cells (*Shu et al., 2022*), we further examined whether rsNS1 oligomerizes in the presence of exogenously added human HDL using XL-MS. The crosslinked rsNS1 alone appeared as higher oligomer bands of dimers, tetramers, and hexamers on the SDS-PAGE analysis (*Figure 4f*, lane 3, *Figure 4—source data 2*), which differed from the crosslinked isNS1 (*Figure 4a*, lane 3). Interestingly, the addition of human HDL and crosslinker resulted in a smearing band ranging from 30 to 150 kDa with the loss of the well-defined monomeric rsNS1 band (*Figure 4f*, lane 4). In the crosslinked rsNS1 in the absence of HDL, we identified 17 NS1:NS1 intramolecular crosslinks at the β-roll and wing domains (*Figure 4f and g*, *Supplementary file 4 and 5*). In the presence of human HDL, we identified 30 ApoA1:ApoA1, 4 NS1:NS1, and 6 NS1:ApoA1 crosslinks (*Supplementary file 4 and 5*). Most crosslinks identified are ApoA1:ApoA1 intramolecular crosslinks in the rsNS1:ApoA1 samples (*Figure 4g*). Only one out of six rsNS1:ApoA1 intermolecular crosslinks was within the 30 Å cutoff distance when mapped to the NS1:ApoA1 cryoEM model (*Figure 4e*). This was in stark contrast with the isNS1 samples where the number of crosslinks between ApoA1:ApoA1, sNS1:sNS1, and sNS1:ApoA1 was similar (*Figure 4b*). Using a sphere with a radius of 30 Å to represent the maximum range between the Cα atoms that can be crosslinked to the β-roll (residue Ser2), we further show that it is improbable to accommodate the ApoA1 at the core of the hexamer (*Figure 4—figure supplement 2*). Overall, this suggests that rsNS1 may form oligomers of dimers but not a complex with HDL under the tested conditions.

## Ab56.2 binds at the C-terminal β-ladder region of NS1

The cryoEM map for isNS1wt:Fab56.2 reliably placed the epitope site of Ab/Fab56.2 at the C-terminal β-ladder region, although the precise identification of the epitope site was hindered by the limited resolution of the cryoEM maps (*Figures 2 and 3*, *Figure 2—figure supplements 1 and 2*, *Figure 3—figure supplement 1*). Attempts to improve the resolution were challenging, so we resorted to using a truncated form of recombinant secreted NS1 consisting of only the C-terminal region (rsNS1c; residues 172–352) to determine Ab56.2 epitope. The rsNS1c fragment of residues 172–352 was previously used for structural studies of dengue (*Edeling et al., 2014*; *Modhiran et al., 2021*) and other flaviviruses, West Nile virus (*Edeling et al., 2014*), Zika virus (*Modhiran et al., 2021*; *Wang et al., 2017*), and Japanese encephalitis virus (*Poonsiri et al., 2018*). The epitope region of Ab56.2 was first confirmed by overlapping 15-mer NS1 peptide competition ELISA for Ab56.2 binding (*Figure 5*, *Table 2*). The greatest competition was observed at the NS1 peptide spanning residues 316–330 out of 301–340 which showed a decrease in OD450 absorbance (*Figure 5b*).

We further applied the hydrogen-deuterium exchange mass spectrometry (HDX-MS) to monitor the conformational dynamics between Fab56.2 and rsNS1c to obtain higher peptide-level resolution for the binding interface. Comparative HDX results revealed global-scale protection against deuterium exchange of rsNS1c in the presence of Fab56.2 (*Figure 6a*, *Figure 6—figure supplement 1a*), indicative of reduced conformational dynamics. Large-scale differences in deuterium exchange were primarily observed across peptides spanning the dimer interface (residues 174–186) and at the tip of the C-terminal β-ladder (residues 286–347) of rsNS1c (*Figure 6b*). Multiple overlapping peptides, spanning residues 286–347 (*Figure 6b*), showed protection against deuterium exchange even at the longer labeling times (100 min) in rsNS1c:Fab56.2 complex (*Figure 6—figure supplement 1a*, *Supplementary file 6*). These results suggest that Fab56.2 bound stably with rsNS1c. Coupled with

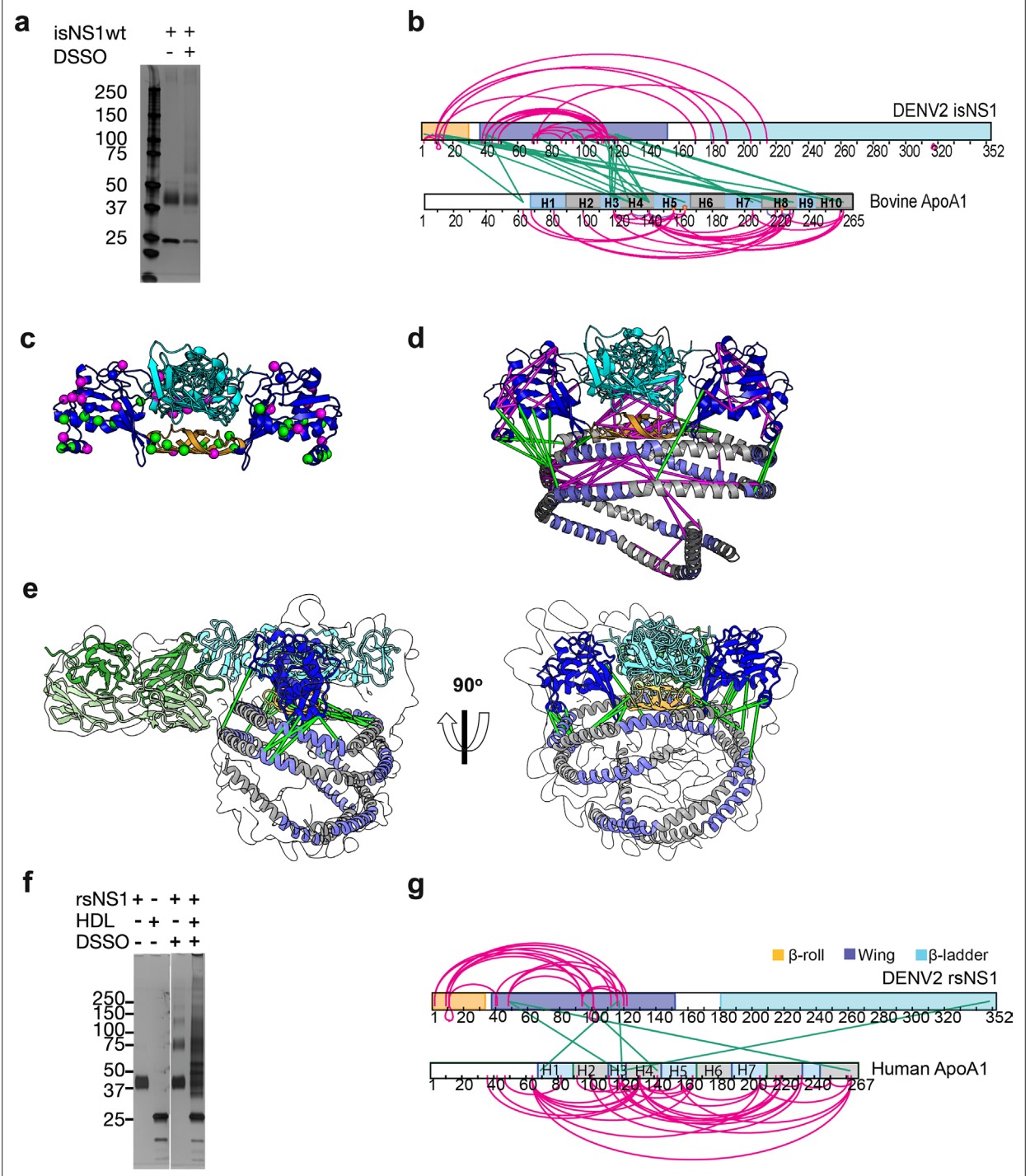

**Figure 4.** Interaction sites of NS1:ApoA1 complex identification by crosslinking mass spectrometry. (**a**) SDS-PAGE analysis of isNS1wt with or without the addition of disuccinimidyl sulfoxide (DSSO) crosslinker. (**b**) The identified crosslinks are visualized on the NS1 and bovine ApoA1 constructs. The intramolecular (NS1:NS1 and ApoA1:ApoA1) crosslinks are in magenta. The intermolecular NS1:ApoA1 crosslinks are in green. (**c**) The isNS1 residues that are involved in NS1:NS1 interactions are visualized on the NS1 dimer model. The NS1 cartoon model is colored by its three domains, namely the β-roll (orange), wing (blue), and β-ladder (cyan) with the intramolecular (magenta) and intermolecular (green) crosslinking sites depicted as spheres. (**d**) The overall model interpretation of NS1:ApoA1 complex within the crosslinker theoretical distance cutoff at <30 Å as depicted. ApoA1 dimer cartoon model with its conserved helices as labeled colored in intervals of gray and light purple. (**e**) The NS1:ApoA1 dimer model with validated crosslinks was fitted into the cryoEM envelope. (**f**) SDS-PAGE analysis of crosslinked rsNS1 alone or with human high-density lipoprotein (HDL) (lanes 3–5). Non-crosslinked rsNS1 and human HDL are the control (lanes 1–2). The crosslinked rsNS1 can be seen in higher oligomers (lane 3). (**g**) Identified crosslinks are mapped on the NS1 and ApoA1 constructs, colored as per panel (**b**).

*Figure 4 continued on next page*

*Figure 4 continued*

The online version of this article includes the following source data and figure supplement(s) for figure 4:

**Source data 1.** Raw and annotated image for the PAGE gel visualized using silver stain.

**Source data 2.** Raw and annotated image for the PAGE gel visualized using silver stain.

**Figure supplement 1.** Crosslinking mass spectrometry of sNS1:ApoA1 complex.

**Figure supplement 1—source data 1.** Raw and annotated image for the western blot analysis (anti-NS1) on an SDS-PAGE gel.

**Figure supplement 1—source data 2.** Raw and annotated image for the western blot analysis (anti-NS1) on a Native gel.

**Figure supplement 2.** NS1 hexamer and dimer-HDL models with residues identified in intra- and intermolecular crosslinking interactions.

the observation that the most significant change in the differential plot was at residues 300–310 (*Figure 6a*), these results collectively indicate that rsNS1c binds stably to Fab56.2 at the primary epitope site of residues 300–310.

Comparative HDX results of rsNS1c-bound and free Fab56.2 revealed protection against deuterium exchange of all three complementarity-determining regions (CDR) of the heavy chain (*Figure 6c*, *Figure 6—figure supplement 1b*) and CDRL2 of the light chain (*Figure 6d*, *Figure 6—figure supplement 1c*). Upon binding to rsNS1c, peptides spanning CDRH3 (residues 89–116) showed the largest magnitude decrease in deuterium exchange at all labeling time points, while CDRH2 showed minor significant differences. In addition, only the CDRL2 loop (residues 49–56) of the light chain of Fab56.2 showed significantly decreased deuterium exchange. Mapping of the deuterium exchange profiles of heavy and light chains onto a model of Fab56.2 (*Figure 6e*) revealed that CDRH1-3 and CDRL2 loops were spatially co-localized to form the paratope site to bind rsNS1c, with the heavy chain being the primary anchor. Collectively, these results indicated that residues 300–310 of NS1 act as the primary epitope site to bind Fab56.2 and Fab56.2 binds stably with rsNS1c.

## Circulating NS1 is in the form of NS1:HDL complex

Lastly, we sought to provide in vivo evidence of NS1 and ApoA1 interaction using sera samples from DENV-infected mice and a human patient. Since we previously showed that DENV2 T164S mutant virus led to more severe disease in the AG129 mouse model (*Chan et al., 2019*), we used the same infection model to obtain pooled mice serum for immunoaffinity purification of sNS1ts as described in *Figure 1*. The mouse sera immunoaffinity purified sNS1ts was detected at ~250 kDa by both anti-ApoA1 and anti-NS1 antibodies in a western blot analysis following separation on a Native-PAGE (*Figure 7a*, *Figure 7—source data 1*), supporting our mass spectrometry finding that ApoA1 is part of the ~250 kDa isNS1 protein complex purified from infected Vero cells (*Figure 1d*). We next examined if the isNS1:HDL complex can be detected in dengue patients using serum samples obtained from the CELADEN clinical trial (*Low et al., 2014*). A primary infection sample with high levels of sNS1 (>15 µg/mL) detected by commercial Platelia NS1 ELISA was selected for this purpose to avoid the confounding factor of the presence of anti-NS1 Abs in the secondary infected patients and to pull-down sufficient sNS1 from the patient serum. The patient serum was first subjected to a pre-clearing step using Protein AG resin to reduce the abundance of human IgGs that may interfere with the downstream analysis. We then subjected the pre-cleared patient serum to immunoprecipitation using commercial polyclonal anti-ApoA1 antibodies to pull down and detect ApoA1 and possibly sNS1. Increasing the anti-ApoA1 antibody from 10 to 50 µg coupled onto Protein AG resin for pull-down resulted in a corresponding increase in the sNS1 amount detected by ELISA (*Figure 7b*), suggesting a specific association between sNS1 and ApoA1. The observation that NS1 interacts with HDL regardless of the source of mammalian ApoA1 is unsurprising, given the high sequence similarity of bovine and mouse HDL to human HDL of approximately 79 and 65%, respectively (*Figure 7*, *Figure 7—figure supplement 1*), and that the structure of ApoA1 is highly conserved (*Malajczuk et al., 2021*). Collectively, despite the limited amount of sNS1 in sera samples from DENV-infected mice or humans, our data points to the association of ApoA1 with sNS1 in vivo.

## Discussion

In this study, by integrating biochemical and biophysical approaches including immunoaffinity purification, cryoEM, XL-MS, and HDX-MS, we have successfully purified and characterized a native

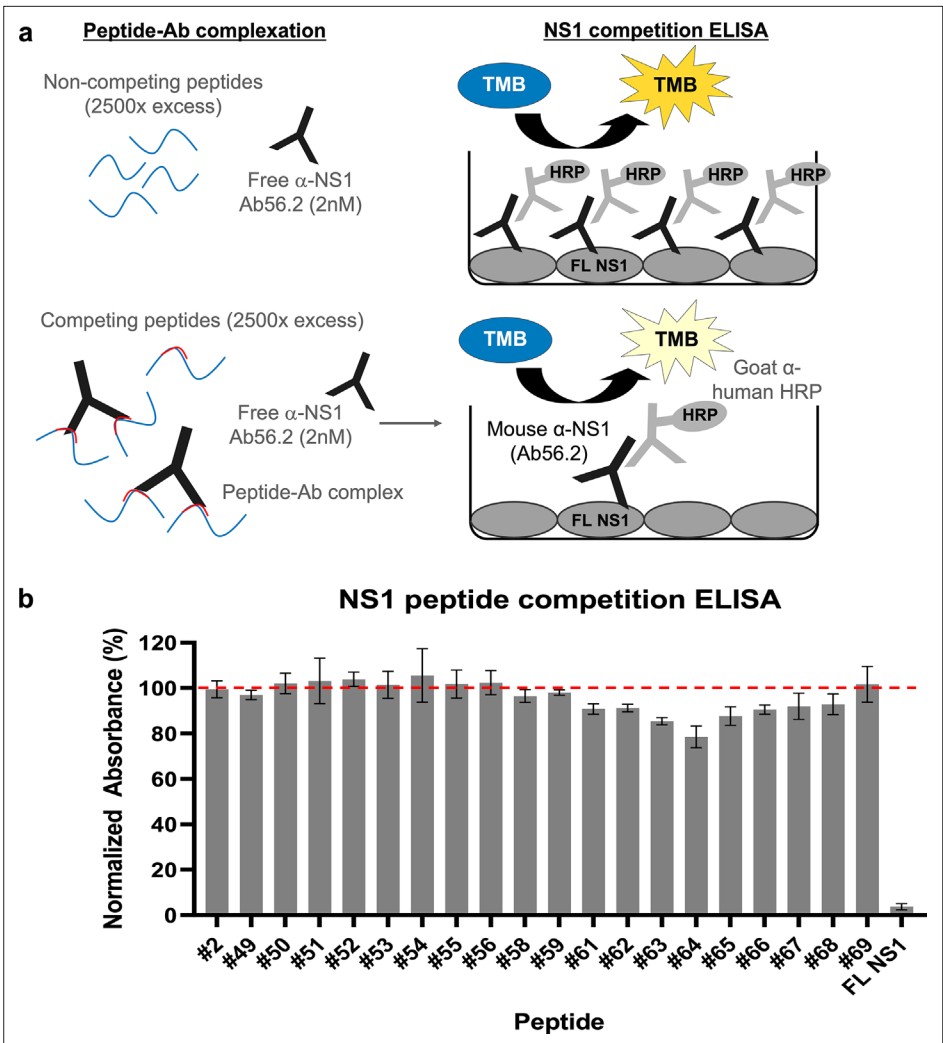

**Figure 5.** Epitope mapping of Ab56.2 by NS1 peptide competition ELISA. (**a**) Schematic of the principle of NS1 peptide competition ELISA to determine the epitope of Ab56.2. NS1 peptides (15-mer, 2500× molar excess) were first incubated with 2 nM of Ab56.2 at room temperature for formation of peptide-Ab complex. The complexation mixture is then added to ELISA plate pre-coated with full length (FL) DENV2 NS1. If the peptides contain epitope that is recognized by Ab56.2, the peptides will compete with the FL NS1 binding to the Ab56.2 in the ELISA plate, resulting in a reduced signal compared to non-competing peptides. (**b**) Normalized absorbance (%) of Ab56.2 to the FL NS1 in the presence of NS1 15-mer peptides. The absorbance reading is normalized to the signal in the absence of the peptides. FL NS1 protein is used as a positive control for the NS1 peptide competition ELISA assay. Results shown are mean ± SD from two independent experiments. Residue sequences are listed in *Table 2*. β-roll residues 6–20 (#2) were used as a non-competing peptide and the red-dashed line represents the threshold for determining the competing peptides.

population of isNS1. We determined the cryoEM structure of a natively secreted NS1 from DENV2-infected cells as a ternary complex of NS1 dimer embedded on a single HDL particle. Our in-solution XL-MS analysis captured interactions within a cutoff distance of 30 Å imposed by the linker-spacer between the β-roll and wing domains of NS1 and helices 3, 5, 8, and 10 of (bovine) ApoA1 – the main protein moiety of HDL. More importantly, we were able to detect NS1:HDL complexes in vivo in both the sera of DENV2-infected mice and in the serum of a DENV1-infected patient, highlighting the biological relevance of the NS1:HDL complexes in dengue infection.

Benfrid et al. (2022) recently reconstituted complexes of rsNS1 dimers bound to the surface of a spheroid human HDL particle in vitro at 1:1 to 3:1 ratio when up to 400 µg/mL of rsNS1 were used *Benfrid et al., 2022*. Similarly, in our studies with infection-derived sNS1 (isNS1), we observed up to two NS1 dimers docked onto an HDL particle (*Figure 1—figure supplement 1c*, *Figures 2 and 3*,

**Table 2.** EDEN2 NS1 peptides used in epitope mapping.

Residues in red are the difference compared to DENV2 16681. Peptides 57 and 60 (aa residues 281–295 and 296–310, respectively) failed the QC check, no peptide made.

| Peptide # | AA residues # | AA residues |
|---|---|---|
| 2 | 6–20 | VSWKNKELKCGSGIF |
| 49 | 241–255 | MIIPKNFAGPVSQHN |
| 50 | 246–260 | NFAGPVSQHNYRPGY |
| 51 | 251–265 | VSQHNYRPGYHTQTA |
| 52 | 256–270 | YRPGYHTQTAGPWHL |
| 53 | 261–275 | HTQTAGPWHLGRLEM |
| 54 | 266–280 | GPWHLGRLEMDFDFC |
| 55 | 271–285 | GRLEMDFDFCEGTTV |
| 56 | 276–290 | DFDFCEGTTVVVTED |
| 58 | 286–300 | VVTEDCGNRGPSLRT |
| 59 | 291–305 | CGNRGPSLRTTTASG |
| 61 | 301–315 | TTASGKLITEWCCRS |
| 62 | 306–320 | KLITEWCCRSCTLPP |
| 63 | 311–325 | WCCRSCTLPPLRYRG |
| 64 | 316–330 | CTLPPLRYRGEDGCW |
| 65 | 321–335 | LRYRGEDGCWYGMEI |
| 66 | 326–340 | EDGCWYGMEIRPLKE |
| 67 | 331–345 | YGMEIRPLKEKEENL |
| 68 | 336–350 | RPLKEKEENLVNSLV |
| 69 | 341–352 | KEENLVNSLVTA |

*Figure 2—figure supplements 1 and 2*, *Figure 3—figure supplement 1*). It is noteworthy that the concentration of isNS1:HDL in our study (~30 µg/mL) is closer to the maximum clinically reported concentration of sNS1 in the sera of DENV patients (50 µg/mL) *Libraty et al., 2002*; *Young et al., 2000*; *Alcon et al., 2002*. Therefore, the ratio of sNS1 to HDL varies depending on the concentration of the proteins used for complex formation.

While our study is limited to only one antibody (56.2) for immunoaffinity purification, our key take-home message in terms of structure is that our data does not support the widely held view that sNS1 is composed of a trimer of NS1 dimers with an ~28 Å wide central channel filled with lipid as established in several seminal publications (*Flamand et al., 1999*; *Gutsche et al., 2011*; *Muller et al., 2012*). In fact, our data confirms the observations reported in *Benfrid et al., 2022*. In the rare instances where trimer of dimers have been experimentally shown by cryoEM of rNS1 structures (*Shu et al., 2022*), only a single lipid molecule can be found in the narrow central channel and unlikely to be reflective of sNS1 circulating in serum. We argue that the amount of lipid associated with infection-derived sNS1 (*Gutsche et al., 2011*; *Chan et al., 2019*) is consistent with NS1 dimer(s) associated with HDL. The interaction with HDL requires an exposed hydrophobic face of NS1 that is incompatible to the hexamer model of sNS1. Indeed, it is agreeable that the oligomeric state of NS1 is dynamic, which may vary depending on the context and environment. This point requires further independent studies.

Our previous functional studies comparing isNS1wt with isNS1ts (T164S) showed that incubating immunoaffinity-purified sNS1 with human PBMCs from three independent human donors triggered the production of proinflammatory cytokines IL6 and TNFα in a concentration-dependent manner and that it was more pronounced with the mutant than the WT (*Chan et al., 2019*). Furthermore, we have provisional transendothelial electrical resistance (TEER) assay data with isNS1 proteins used

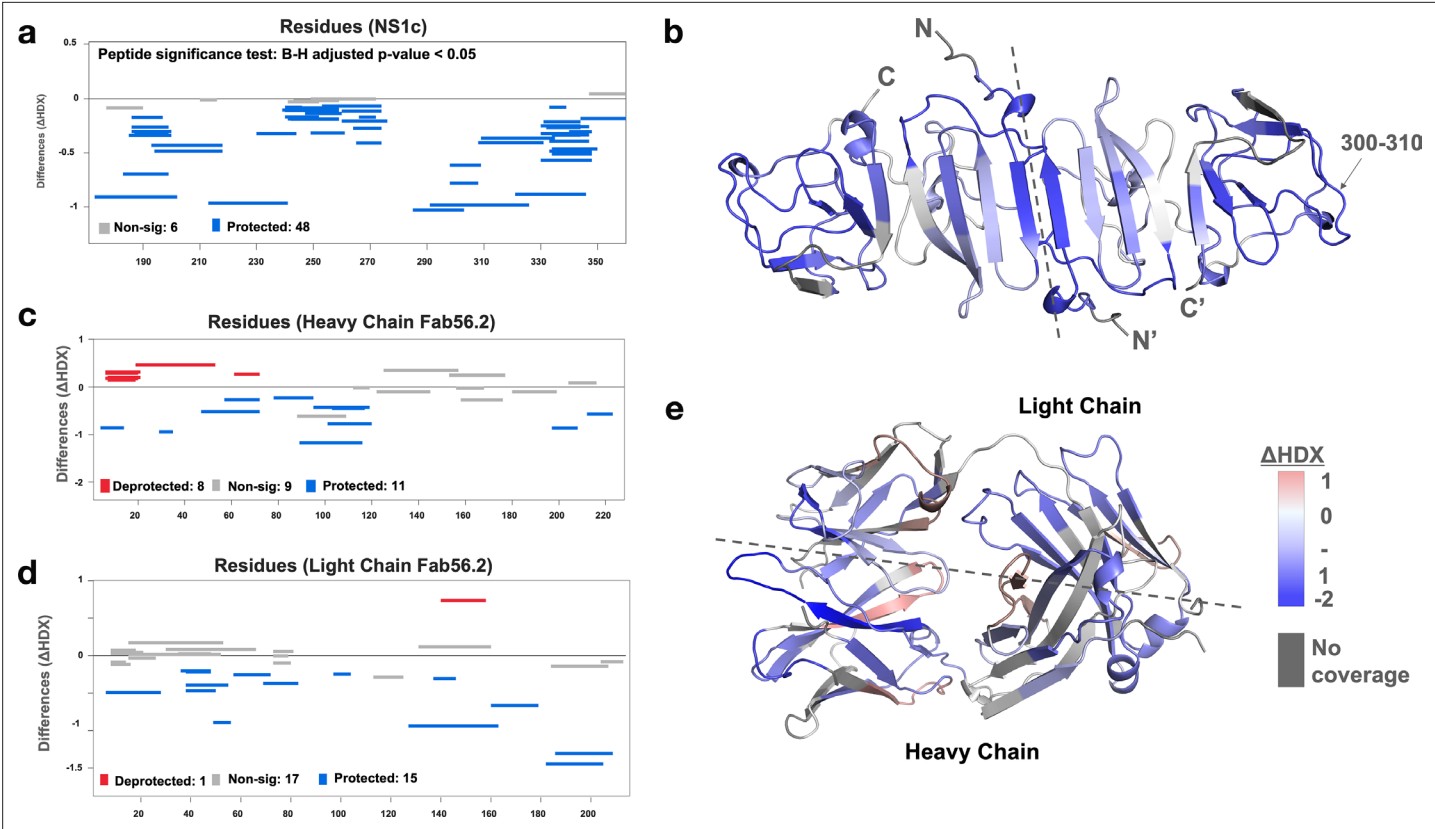

**Figure 6.** Interaction interface of rsNS1c-Fab562 complex characterized by hydrogen-deuterium exchange mass spectrometry (HDX-MS). (**a**) Woods differential plot showing the differences in deuterium exchange (y-axis) between Fab56.2-bound and free rsNS1c protein across various residues (x-axis) at 10 min labeling timepoint. Negative differences indicate protection against deuterium exchange across rsNS1c peptides in the presence of Fab56.2 compared to free rsNS1c. A p-value<0.05 is considered as significance threshold, which identified 6 nonsignificant peptides (gray lines) and 48 protected peptides (blue lines). (**b**) Cartoon representation of rsNS1c dimer model showing differences in deuterium exchange at 10 min labeling as indicated in key. Dashed line distinguishes the two monomers in the dimer. Peptide spanning residues 300–310 showing the highest protection are indicated. (**c, d**) Woods differential plot comparing the differences in deuterium exchange of Fab56.2 in the presence and absence of rsNS1c for various peptides of (**c**) heavy chain and (**d**) light chain. A p-value<0.05 is considered as significance threshold, which identified deprotected (red lines), nonsignificant, and protected peptides as indicated. (**e**) Cartoon representation of Fab56.2 model showing deuterium exchange differences at 10 min mapped for Fab56.2-rsNS1c complex as per key. Plots were generated using Deuteros 2.0, while cartoon structures were generated using PyMoL.

The online version of this article includes the following figure supplement(s) for figure 6:

**Figure supplement 1.** Heat map identified specific residues associated with rsNS1c-Fab562 complex.

in the structural studies with human umbilical vascular endothelial cells (hUVEC), which show that acid-eluted isNS1 proteins in this study induces endothelial hyperpermeability as shown in previous studies (*Puerta-Guardo et al., 2019*). The presence of a significant population of free isNS1ts dimers in complex with Fab56.2 and fewer crosslinks identified between isNS1ts and ApoA1 point toward the possibility that isNS1ts is more dynamic in its association with host proteins than isNS1wt. The disease severity and increased complement protein expression in AG129 mice liver infected with NS1T164S carrying virus compared with WT that was observed in our previous study (*Chan et al., 2019*) can be ascribed to weakly bound mutant NS1 with fast on/off rate with HDL being transported to the liver where specific receptors bind to free sNS1 and interact with effector proteins such as complement to drive inflammation and associated pathology. This notion also requires further validation with other mutations that can impact sNS1 –HDL interaction.

Overall, our work provided a refined understanding of the native structure of sNS1 in DENV pathogenesis. The functional role of HDL-bound NS1 protein in disease is now a topic of contention but one that will be resolved by more data obtained with infection-derived sNS1 by scientists in the field. Contrary to the known protective roles of HDL such as against endotoxic shock caused by lipopolysaccharides (*Trinder et al., 2020*; *Tall, 2021*; *Harsløf et al., 2023*), Benfrid et al. (2022) showed that

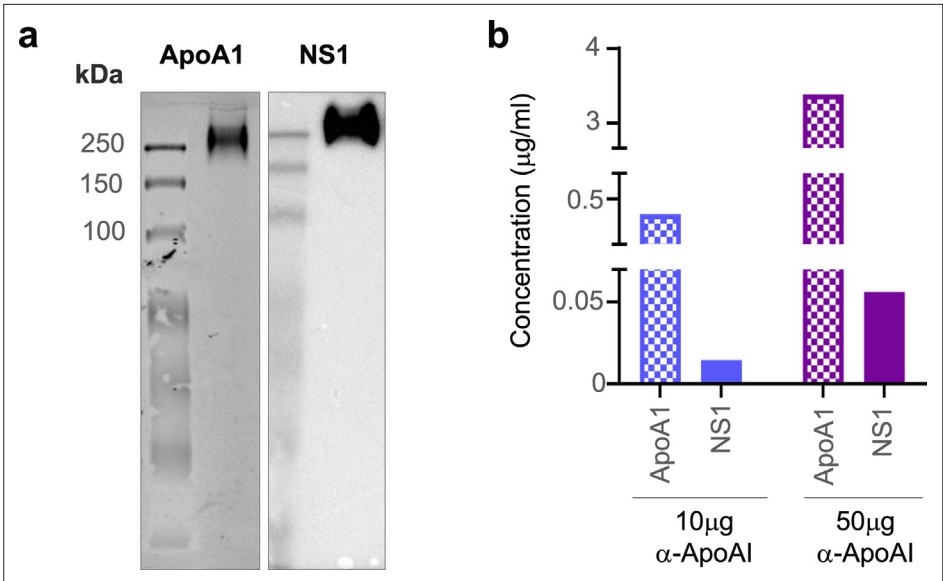

**Figure 7.** sNS1 associates with ApoA1 in dengue virus (DENV)-infected mouse and human serum. (**a**) AG129 mice (n = 10) were infected with DENV2 NS1 T164S mutant virus and the pooled infected sera collected on day 4 post-infection was subjected to sNS1 immunoaffinity purification using anti-NS1 56.2 coupled resin as in *Figure 1—figure supplement 1*. 2 mg of the purified eluate was then subjected to western blot analysis after separation and transfer from Native-PAGE for detection of ApoA1 and NS1. ApoA1 was detected using the mouse monoclonal anti-ApoA1 clone 513 (Invitrogen, MIA1404) (left panel) and the oligomeric NS1 was detected using anti-NS1 56.2 IgG clone (right panel). (**b**) Protein AG resin (Pierce) pre-cleared DENV1-infected patient serum (n = 1) from the CELADEN trial (*Low et al., 2014*) was immunoprecipitated with 10 or 50 mg of rabbit polyclonal anti-ApoA1 antibody (Biorbyt, orb10643) to detect association between ApoA1 and NS1 by ELISA. The amount of ApoA1 and NS1 in the immunoprecipitated sample was determined by human ApoA1 (Abcam, ab189576) and Platelia NS1 Ag (Bio-Rad) ELISAs.

The online version of this article includes the following source data and figure supplement(s) for figure 7:

**Source data 1.** Raw and annotated image for the western blot analysis (anti-NS1 and anti-apoA1) on a Native gel.

**Figure supplement 1.** Pairwise sequence alignment of bovine and human ApoA1.

---

sNS1:HDL complex triggers proinflammatory signals when tested with primary human macrophages (*Benfrid et al., 2022*). The presence of sNS1:HDL complexes as a predominant population of sNS1 in our in vitro infection model corroborates with a previous report of an interaction between insect-derived recombinant NS1 and the classical HDL/ApoA1-binding receptor known as SR-BI (*Alcalá et al., 2022*), which is primarily expressed on hepatocytes (*Acton et al., 1996*). This interaction in hepatocytes has been suggested to trigger endocytosis of sNS1 and subsequent downstream effects including vascular permeability, which is a characteristic feature of severe dengue (*Alcalá et al., 2022*; *Alcon-LePoder et al., 2005*; *Wang et al., 2019*). Internalization of sNS1 is critical for increased virus production in hepatocytes (*Alcon-LePoder et al., 2005*) and NS1-mediated permeability in endothelial cells (*Wang et al., 2019*). Furthermore, NS1:HDL complexes have been shown to induce the production of pro-inflammatory cytokines in macrophages and their presence has been detected in the sera of hospitalized DENV patients *Benfrid et al., 2022*. Yet, this hypothesis is confounded by other reports. While rsNS1 has been shown to trigger a TLR4-mediated release of proinflammatory cytokines to induce vascular leakage (*Modhiran et al., 2015*; *Coelho et al., 2021*), Coelho et al. (2021) further showed that lipid-free recombinant ApoA1 neutralizes the proinflammatory effects caused by insect-derived recombinant intracellular NS1 (*Coelho et al., 2021*). Furthermore, exogenous administration of purified HEK-293-derived NS1 failed to worsen in vivo vascular leakage in sublethally infected mice (*Lee et al., 2020*). This underscores the need for physiologically relevant studies to reconcile these discrepancies and gain a comprehensive understanding of the role of NS1-HDL complex in DENV pathogenesis.

In conclusion, despite the study limitations that our isNS1wt and isNS1ts were immunoaffinity purified with one specific monoclonal antibody (Ab56.2), our structural model, obtained by cryoEM and verified by XL-MS, of infection-derived sNS1 shows a NS1 dimer embedded on a single HDL particle composed of ApoA1 dimer. Future investigation of this work using infected mouse sera from other DENV strains and different flaviviruses would inform the structural and functional relevance of the HDL-bound NS1 protein complex among flaviviral NS1 as well as provide a basis for the development of NS1-directed therapeutics.

## Materials and methods

### Cells, viruses, and antibodies

Vero cells (green African monkey kidney epithelial cells, ATCC) were cultured in DMEM containing 4.5 g/L glucose (Gibco) supplemented with 10% (v/v) fetal bovine serum (FBS) and 1% (v/v) penicillin-streptomycin (P/S) at 37°C in 5% $CO_2$. Expi293F cells were cultured in serum-free Expi293 Expression Medium at 37°C in 8% $CO_2$. The infectious clone-derived DENV2 WT (GenBank accession: EU081177) and NS1:T164S mutant viruses used in this study were described previously (*Chan et al., 2019*). The anti-NS1 56.2 IgG antibody (Ab56.2) was obtained from the hybridoma cell culture as described previously (*Rozen-Gagnon et al., 2012*). Briefly, the Ab56.2 hybridoma cells were grown in PFHM II medium (Gibco) at 37°C in 5% $CO_2$. Culture supernatants were collected every 4 d, clarified, and filtered through 0.2 µm filter membrane. Ab56.2 was purified from the supernatant through a Protein G HiTrap column (GE Healthcare) using the AKTA purification system (Cytiva). The bound Ab was eluted from the Protein G column using 0.1 M glycine (pH 2.7), neutralized with 1 M Tris-HCl (pH 9.0), and dialyzed with PBS for storage at –30°C until use. Cell lines were tested negative for mycoplasma contamination.

### Generation and purification of sNS1 from virus infection

Vero cells in 20 T175 flasks were infected at a multiplicity of 0.1 with DENV2 WT and T164S viruses for 1 hr in serum-free DMEM and subsequently replaced with 25 mL of DMEM supplemented with 2% FBS per flask. The flasks were then incubated for 72 hr at 37°C in 5% $CO_2$. The crude supernatant (500 mL) was then harvested, clarified, and filtered through a 0.2 µm filter membrane (Nalgene, Thermo Fisher), followed by supplementation with 0.05% sodium azide and cOmplete EDTA-free protease inhibitor cocktail (Roche, Sigma-Aldrich). The crude supernatant was then concentrated tenfold volume-wise using a Vivaflow 200 cassette with a 100 kDa MWCO (Sartorius) attached to a peristaltic pump (Cole Parmer). 5 mL of Aminolink resin (Thermo Fisher) with immobilized Ab56.2 was then added to the crude supernatant and allowed end-over-end rotation overnight at 4°C for batch immunoaffinity purification.

The slurry was then poured into a Econo-Pac Chromatography Column (Bio-Rad), washed with filtered PBS (pH 7.4) for at least 10 column volumes and eluted with 3 column volumes of 0.1 M glycine (pH 2.7), and immediately neutralized with 1 M Tris-HCl (pH 9.0). The eluted protein was dialyzed against PBS (pH 7.4) overnight at 4°C and subsequently concentrated using a 100 kDa MWCO Amicon ultracentrifugal unit (Millipore, Merck). The total protein concentration was determined using Bradford assay (Bio-Rad) and the Platelia NS1 ELISA kit (Bio-Rad). Purified baculovirus-derived sNS1, as described previously (*Rozen-Gagnon et al., 2012*), was used to generate a standard curve ranging from 20 ng/mL to 312.5 pg/mL for the NS1 ELISA. Protein quality was assessed on a 4–20% polyacrylamide SDS gel and a 10% Native polyacrylamide gel and transferred to PVDF membranes for western blot analysis against NS1 (Ab56.2) and ApoA1 (Biorbyt, orb10643). Protein purity was assessed on a 4–20% polyacrylamide SDS gel and stained using Coomassie blue (0.2% Coomassie blue, 7.5% acetic acid, 50% methanol). The Precision Plus Protein Dual Color Standard (Bio-Rad) was used as the ladder for all protein gels in this work. Western blots and Coomassie blue-stained gels were visualized with a Chemidoc Imager (Bio-Rad). The purified protein was stored at –80°C until use.

### Cloning and plasmid preparation of recombinant His-tagged NS1c construct

The DENV 2 NS1c fragment was amplified from the full-length DENV2 3295 infectious clone plasmid DNA as previously described (*Chan et al., 2019*) using the forward primer that also includes a

receptor-type tyrosine-protein phosphatase S (PTPRS) signal peptide: 5'- GGGTTGCGTAGCTGAA ACCGGTAAAGAAAGGCAGGATGTATCTTGT-3' and reverse primer: 5'- GGTGGTGCTTGGTACC GGCTGTGACCAAAGAGTTGACC-3'. The fragment was then cloned into the 6xHis sequence-containing pHL-sec vector at XbaI/KpnI cloning sites. The plasmid was then amplified in XL1-blue *Escherichia coli* (Agilent Technologies), and plasmid preparation was performed using the Plasmid DNA Maxiprep kit (Thermo Fisher).

## Generation and purification of recombinant His-tagged rsNS1c
Expi293F cells were transfected at a cell density of $3 \times 10^6$ viable cells/mL and with 1 µg of the NS1c plasmid DNA/mL culture volume using the ExpiFectamine 293 Transfection kit (Gibco, Thermo Fisher) according to the manufacturer's instructions. On day 5 post transfection, the cells were spun down and the supernatant containing the expressed His-tagged NS1c was harvested, clarified, and filtered using a 0.2 µm filter membrane (Millipore, Merck). The His-tagged NS1c was then purified from the supernatant using a 1 mL HisTrap column (Cytiva), concentrated using a 50 kDa MWCO Amicon concentrator (Millipore, Merck), and stored in –80°C until use. Protein yield and quality were assessed on a 4–20% reducing SDS-PAGE and stained with Coomassie blue, as described in 'Generation and purification of sNS1 from virus infection'.

## Generation and purification of anti-NS1 56.2 Fab
1.8 mg of the purified Ab56.2 was subjected to papain digestion using the immobilized papain resin (Thermo Fisher) at an enzyme:substrate ratio of 1:160 according to the manufacturer's instructions. The IgG-papain mixture was incubated at 37°C for 3.5 hr followed by removal of the immobilized papain resin to stop the digestion. The papain IgG digest was buffer-exchanged to PBS using PD10 Sephadex G25 column (GE Healthcare). The resulting Fab (Fab56.2) was then purified from the crude digest by Protein A HiTrap column (GE Healthcare) using the AKTA purification system (Cytiva), concentrated using 3 kDa MWCO Amicon concentrator (Millipore, Merck) and stored at –30°C until use.

## Analytical size-exclusion chromatography for isNS1, Fab, and isNS1:Fab complexes
10 µg of isNS1wt or isNS1ts was complexed with papain-digested Fab56.2 or Ab56.2 at a protein:Fab or protein:Ab molar ratio of 1:5 by incubation for at least 2 hr on ice. 10 µg of isNS1wt, isNS1ts, or papain-digested Fab56.2 as well as the isNS1wt: or isNS1ts:Fab56.2 complex was then independently injected onto a Superdex 200 increase 3.2/300 GL column (GE Healthcare) connected to the AKTA purification system (Cytiva) in PBS (pH 7.4) at a constant flow rate of 0.075 mL/min. Chromatograms were analyzed on Unicorn 7 and replotted on OriginPro, version 2021b.

## Negative stain microscopy and data processing
3 µL of protein sample at concentration of 0.01 mg/mL were spotted on glow-discharged carbon grids, contrasted with 2% uranyl acetate, and imaged with a FEI Tecnai T12 microscope equipped with an Eagle 4 megapixel CCD camera (Thermo Fisher, USA). Twenty-four micrographs were manually collected and processed using Scipion3 (*de la Rosa-Trevín et al., 2016*). CTF estimation, manual, and auto-picking were done using xmipp3 (*de la Rosa-Trevín et al., 2013*), resulting in a total of 3914 particles. This is followed by particle extraction and 2D classification using Relion (*Zivanov et al., 2018*).

## CryoEM grid preparation and microscopy
QuantiFoil or UltrAuFoil R1.2/1.3 gold 300 mesh grid was covered with a graphene layer (Graphenea) following an adapted protocol (*Han et al., 2020*) and glow-discharged for 10 s at low energy (Harrick Basic Plasma Cleaner) before use. 2.5 µL of protein sample at a concentration of 0.35 mg/mL was applied to the grids, blotted for 3 s with blot force 1, and plunge-frozen in liquid ethane using Vitrobot (Thermo Fisher Scientific). CryoEM grids for isNS1, isNS1wt:Ab562, and isNS1wt:Fab562 were imaged using EPU v2.14 on a 300 kV TEM Titan Krios with Fringe-free imaging and aberration-free image shift (Thermo Fisher Scientific) and equipped with a K2 direct electron detector (Gatan). Data collection was performed at a nominal magnification of ×165,000 with a physical pixel size of 0.85 Å. A GIF Quantum Energy Filter with a slit width of 20 eV was used. Micrographs were dose-fractioned into

40–50 frames with a total exposure dose of 53–70 e- per Å$^2$. The isNS1ts mutant in complex with Fab56.2 dataset was collected on a Titan Krios equipped with a K3 direct electron detector (Gatan) using SerialEM v4.06 (*Mastronarde, 2005*). Data collection was performed at a nominal magnification of ×105,000 with a physical pixel size of 0.858 Å. A GIF Quantum Energy Filter with a slit width of 20 eV was used. Micrographs were dose-fractioned into 40 frames with a total exposure dose of 53 e- per Å$^2$. The data collection parameters are summarized in *Table 1*.

## CryoEM image processing and model fitting

Collected movies were imported to Cryosparc v3.3 or later (*Punjani et al., 2017*) and processed similarly for all datasets starting with Patch-Based Motion Correction and CTF estimation. Micrographs with CTF-estimated maximum resolution better than 4.5 Å were selected for particle picking starting with blob-based picker followed by a combination of Template Picker, Topaz (*Bepler et al., 2019*), and crYOLO v1.7.6 (*Wagner et al., 2019*) after initial rounds of 2D classification using extracted particles that were Fourier cropped by four times had been performed. Particles were extracted at a box size at 300 pixels for isNS1ts (*Figure 1—figure supplement 4*), 352 pixels for isNS1wt:Ab562 (*Figure 2—figure supplement 1*), and 416 pixels for isNS1wt:Fab562 (*Figure 2—figure supplement 2*) and isNS1ts:Fab562 (*Figure 3—figure supplement 1*). Additional two rounds of 2D classification without downsampling were performed before the selected particles were subjected to ab initio reconstruction for three classes. All resulting classes were refined respectively with no symmetry (C1) using heterogeneous refinement for further cleaning followed by non-uniform refinement (*Punjani et al., 2020*). In general, all the datasets led to low-resolution maps and the class maps of isNS1wt:Ab562, isNS1wt:Fab562, and isNS1ts:Fab562 could be fitted with the NS1, Fab56.2, and apoA1 models generated using AlphaFold2 (*Jumper et al., 2021*; *Mirdita et al., 2022*; *Figures 2d and 3d and e*). The final map used for isNS1wt:Ab562 (*Figure 2—figure supplement 1*) for the overall shape and size is derived from heterogeneous refinement as the non-uniform refinement resulted in badly connected maps. Preferred orientation is most apparent for the class maps of isNS1ts:Fab562 (*Figure 3—figure supplement 1*). Further local refinement was attempted for selected classes of isNS1wt:Fab562 (*Figure 2—figure supplement 2*) and isNS1ts:Fab562 (*Figure 3—figure supplement 1*) datasets to improve final maps; however, there was no visible improvement. Histogram and directional Fourier Shell Correlation (FSC) plots were generated for selected classes of isNS1wt:Ab562 (*Figure 2—figure supplement 1*), isNS1wt:Fab562 (*Figure 2—figure supplement 2*), and isNS1ts:Fab562 (*Figure 3—figure supplement 1*) using the remote 3DFSC processing server (*Tan et al., 2017*) for better estimation of the map resolution and directional anisotropy. Map fitting and figures representing the map and model features were performed using UCSF ChimeraX (*Pettersen et al., 2021*). The data collection statistics are summarized in *Table 1*. Data processing is also outlined in *Figure 2—figure supplements 1 and 2* and *Figure 3—figure supplement 1*.

## In-gel protein identification and quantification by LC-MS

5 µg of isNS1wt and isNS1ts were separated on a 12% polyacrylamide SDS gel and a 10% Native polyacrylamide gel and subjected to gel electrophoresis. The gel was then rinsed with MilliQ water before staining with Coomassie blue stain (0.2% Coomassie blue R-250, 7.5% acetic acid, 50% methanol) for 30 min with gentle agitation at room temperature. The gel was then destained with Coomassie blue destaining solution (40% methanol, 10% acetic acid) and rinsed with MilliQ water before visualizing on a Chemidoc Imager (Bio-Rad).

The destained gel bands were excised and cut into smaller pieces (approximate 1 mm × 1 mm size). The gel pieces were washed with 1 mL of wash buffer 1 (50% ethanol, 50 mM triethylbicarbonate [TEAB] pH 8.5) (Sigma) by gentle agitation at room temperature for 5 min. The process was repeated twice. The gel pieces were washed twice with 1 mL of wash buffer 2 (100 % ethanol) for 5 min. The ethanol was removed, and the gel pieces were resuspended in 100 µL 100 mM TEAB pH 8.5. The samples were subsequently reduced with 10 mM Tris(2-carboxyethyl)phosphine hydrochloride (TCEP) (Sigma) by incubating with 20 min at 25°C shaking and were alkylated with 55 mM chloroacetamide (CAA) in the dark for 20 minutes at 25°C shaking. The proteins were digested by incubating with 0.75 µg of trypsin (Thermo Fisher Scientific) at 25°C overnight. The supernatant was removed, and the digested peptides were extracted using 100 µL of 30% acetonitrile (ACN) with 3% formic acid (FA) and 100 µL of 80% ACN with 3% FA. Each extraction step was repeated twice. The digested peptides

were dried and resuspended in 20 µL A*buffer (2% ACN, 0.06% trifluoroacetic acid, 0.5% acetic acid), and 2 µL of samples were injected to LC-MS for protein identification and quantification. The peptides were separated on EasySpray (Thermo Fisher Scientific) column with 75 µm inner diameter, 50 mm length with particle size 2 µm and a pore size of 100 Å over a 45 min gradient starting with buffer A containing 0.5% (v/v) FA to buffer B containing 95% ACN with 0.5% (v/v) FA with nanoflow LC pump Easy nLC1200 at a flow rate of 300 nL/min. The data was acquired on Thermo Fisher Orbitrap HF-X mass analyzer or Thermo Fisher Orbitrap Lumos mass analyzer in a data-dependent acquisition mode with each MS scan followed by MS/MS scans. Each duty cycle is 2.5 s long with MS scan at 60,000 m/z resolution followed by MS/MS scans at 7500 m/z resolution with AGC target of $4 \times 10^4$ and dynamic exclusion of 30 s. Raw files were analyzed and quantified using proteome discoverer (PD), version 2.4 (Thermo Fisher Scientific). The proteins were identified using MASCOT search engine using UniProt Bovine Proteome and NS1 protein sequence from DENV2 Singapore clinical isolate with GenBank accession number EU081177.1 with the following parameters, trypsin as the digesting enzyme with maximum two mis-cleavages. The oxidation of methionines and acetylation of protein N-termini, deamidation, were set as variable modifications and carbamidomethylation of cysteines was set as the static modification. The MS tolerance was set at 15 ppm, MS/MS tolerance at 0.08 Da. The precursor ions were quantified using both unique + razor peptides.

## Crosslinking mass spectrometry

Purified isNS1wt complex and isNS1ts were independently crosslinked using disuccinimidyl sulfoxide (DSSO) (Thermo Scientific) in PBS according to the manufacturer's instructions. Briefly, 2.4 µM (or 0.6 mg/mL) of total protein in the purified isNS1wt and 5.8 µM (or 1.45 mg/mL) of total protein in the purified isNS1ts were crosslinked with DSSO in an isNS1:DSSO molar ratio of 1:100 for 45 min at room temperature. The crosslinking reaction was then quenched with 20 mM Tris-HCl (pH 8.0) for 15 min at room temperature. The crosslinked proteins were assessed on a 4–20% reducing SDS-PAGE before subjecting to a silver stain according to the manufacturer's instructions (Pierce) and a western blot using the anti-NS1 antibody Ab56.2.

Commercial DENV2 sNS1 (2.7 µM or 0.675 mg/mL) (Native Antigen Company) and commercial purified HDL (7.1 mM or 2.3 mg/mL) (Innovative Research) in PBS were allowed to associate by incubation on ice for 2 hr in a molar ratio of 1:1. DSSO crosslinker was then added in 100-fold molar excess. Crosslinking was allowed to proceed for 45 min at room temperature before quenching with 20 mM Tris-HCl (pH 8.0) for 15 min at room temperature. The crosslinked products were assessed on a 4–20% reducing SDS-PAGE before subjecting to a silver stain according to the manufacturer's instructions (Pierce) and a western blot using the anti-NS1 antibody Ab56.2.

The crosslinked proteins were resuspended in 45 µL of 100 mM TEAB, reduced and alkylated with 10 mM of TCEP and 55 mM of CAA. The proteins were digested by incubation with trypsin (Thermo Fisher Scientific) overnight at 25°C. The digested peptides were acidified and desalted on stage tips packed with 3M Empore C18 disc. The peptides were eluted with elution buffer (80% ACN, 0.5% acetic acid) and dried. The desalted peptides were resuspended in 10 µL of A*buffer. 1 µg of digested peptides were separated on an EasySpray (Thermo Fisher Scientific) column with 75 µm inner diameter, 50 mm length with particle size 2 µm and a pore size of 100 Å over a 75 min gradient starting with buffer A containing 0.5% (v/v) FA to buffer B containing 95% ACN with 0.5% (v/v) FA with nanoflow LC pump Easy nLC1200 at a flow rate of 300 nL/min. The data was acquired on an Orbitrap Fusion Lumos mass spectrometer (Thermo Fischer Scientific) with CID-MS2/HCD-MS3 fragmentation. MS1 scans were performed in the Orbitrap with 60,000 m/z resolution, AGC target of 400,000, and maximum injection time of 35 ms. Precursor ions with positive charge state 3–8 were selected for MS2 fragmentation with dynamic exclusion of 60 s after 1 scan. MS2 scans were acquired in the Orbitrap with 30,000 m/z resolution, AGC target of 50,000, maximum injection time of 50 ms, and CID collision energy of 30%. Targeted mass difference of 31.9721 m/z (signature peak of MS-cleaved DSSO) was selected for MS3 acquisition in the Ion Trap with rapid scan rate, AGC target of 10,000, maximum injection time of 40 ms, and HCD collision energy of 30%. Raw files were searched using Metamorpheus version 0.0.320 (*Lu et al., 2018*), against a fasta database containing the amino acid sequences of DENV2 NS1, and bovine ApoA1 for isNS1wt and isNS1ts samples and human ApoA1 for rsNS1 samples, with a calibration task before searching for crosslinked peptides. Calibrate task was performed with precursor mass tolerance of 10 ppm and product mass tolerance of 20 ppm. Crosslink

search was performed for DSSO crosslinks at K, S, T, Y amino acid sites, with CID MS2 dissociation type and HCD MS3 child scan dissociation. Up to three missed cleavages were allowed and protease was set to trypsin, with precursor mass tolerance of 10 ppm and product mass tolerance of 20 ppm. Fixed modifications were set for carbamidomethylation (C), while variable modifications were set for oxidation (M), deamidation (N,Q), and for DSSO (K,S,T,Y, protein N-terminus). Resultant crosslinked peptides were filtered for q-value ≤ 0.01 (corresponds to 1% false discovery rate). Crosslink sequence representation figures were made using xiVIEW (*Graham et al., 2019*) while crosslinks were mapped to structures using PyXlinkViewer (*Bullock et al., 2018*) in PyMOL (Schrödinger, LLC.) and XL Mapping and Analysis (XMAS) (*Lagerwaard et al., 2022*) with UCSF ChimeraX (*Pettersen et al., 2021*).

## Immunoprecipitation of ApoA1-sNS1 complex from DENV-infected patient serum

To demonstrate the physiological relevance of the sNS1:ApoA1 interaction in vivo, a co-immunoprecipitation of sNS1:ApoA1 was performed on a DENV1 primary infected patient serum obtained from the Celgosivir clinical trial conducted in Singapore (CIRB Ref: 2012/025/E) that contains ~30 µg/mL sNS1 quantified by Platelia NS1 Capture ELISA (Bio-Rad) (*Low et al., 2014*). 75 µL of AG agarose beads was pre-coupled with either 10 µg or 50 µg of rabbit polyclonal ApoA1 Ab (Biorbyt, orb10643). Prior to immunoprecipitation, the patient serum sample was diluted 2× with PBS and subjected to three rounds of Protein AG agarose beads (Pierce) pre-clearing to remove the highly abundant IgGs in the serum. The Ab pre-coupled AG beads were then added to pre-cleared serum sample and incubated overnight with end-to-end rotation at 4°C. The sample-bound AG beads were washed four times with PBS-T (0.1% Tween 20 [v/v]) followed by addition of 60 µL of 0.1 M glycine (pH 2.7) to elute the sample from the resin that was subsequently neutralized with 1 M Tris-HCl (pH 9.0). The ApoA1 and sNS1 in the eluted sample were subjected to quantification by ELISAs (commercial human ApoA1 ELISA, Abcam; Platelia NS1 Capture ELISA, Bio-Rad). A human naïve serum (obtained accordance with the National University of Singapore IRB approval B-12-227) that served as a negative control was processed and analyzed similarly as the patient serum.

## Immunoaffinity purification of sNS1 from infected mouse serum

Sv/129 mice deficient of type I and II IFN receptor (AG129) purchased from B&K Universal (UK) were housed in BSL-2 animal facility in Duke-NUS, Singapore. All animal experiments (protocol 2021/SHS/1646) were approved by the Institutional Animal Care and Use Committee at Singapore Health Services and conformed to the National Institutes of Health (NIH) guidelines and public law. 10 AG129 (12–14-week-old male) mice were pre-injected with 50 µg 4G2 antibody (ATCC, HB-112) intravenously (i.v.) prior to intravenous inoculation of $2 \times 10^7$ pfu of DENV2 T164S mutant virus (*Chan et al., 2019*). The mice were sacrificed on day 4 post-infection by $CO_2$ inhalation for blood collection from the postcaval vein. A total of 6 mL serum was collected, concentrated, and buffer-exchanged with PBS using 100 kDa cutoff Amicon concentrator (Millipore). Finally, ~5 mL of the buffer-exchanged serum was incubated with 1 mL of the anti-NS1 56.2 coupled resin for immunoaffinity purification as described (see 'Generation and purification of sNS1 from virus infection'). The elutes were concentrated using a 30 kDa Amicon concentrator and subjected to ELISA quantification using commercial Platelia Dengue NS1 Ag kit (Bio-Rad) and western blot analysis after separation by Native-PAGE.

## NS1 peptide competition ELISA

The epitope of anti-NS1 Ab56.2 was determined using peptide competition ELISA as previously described (*Moreland et al., 2010*). A schematic on the principle of NS1 peptide competition ELISA is shown in *Figure 5*. An array of overlapping 15-mer peptides spanning DENV2 NS1 C-terminal residues 241–352 (GenBank accession: EU081177, *Table 2*) were synthesized and purchased from GL Biochem (Shanghai) Ltd. 2 nM of Ab56.2 was preincubated with 5 mM (2500× molar excess) for 30 min at room temperature before transferring to an immunoplate pre-coated with DENV2 full-length NS1 (5 mg/mL in 0.1 M NaHCO₃ buffer at pH 9.6). The immunoplate with the peptide-Ab mix was incubated at room temperature for 10 min followed by washing with PBS-T (0.1% v/v Tween 20). For assay controls, a 15-mer NS1 peptide spanning the β-roll residues 6–20 was used as a non-competing peptide while the full-length DENV2 NS1 was used as a competing reagent. Bound Ab was

detected with an anti-mouse HRP, and absorbance readings at 450 nm were measured in duplicates. Results were tabulated as mean percentage absorbance reading normalized to the Ab only control (no peptides).

## Hydrogen-deuterium exchange mass spectrometry (HDXMS)

HDXMS was used to map the epitope and paratope sites of NS1c and Fab562 complexation, respectively. HDX was performed for NS1c and Fab562 individual purified proteins, and NS1c-Fab562 complex obtained from size-exclusion chromatography elution. To initiate the hydrogen-deuterium exchange reaction, ~75 pmol of the protein samples was diluted with phosphate-buffered saline prepared in deuterium oxide (D2O, Cambridge Isotopes) to achieve a 90% final deuteration. The deuterium exchange was carried out by incubating the proteins for 1, 10, and 100 min labeling time-points at 25°C. After the desired labeling time, the reaction was stopped by lowering the pH to 2.5, temperature to 0°C using a chilled quench solution (1 M guanidinium hydrochloride, 0.1 M TCEP). For reference, non-deuterated controls were also carried out by diluting the protein samples in aqueous PBS, followed by quench solution and mass spectrometry analysis.

Each quenched sample was subjected to 3 min proteolytic digestion using an immobilized pepsin cartridge (Enzymate, Waters, USA) maintained at 12°C and built into an HDX sample manager coupled to nanoACQUITY M-class UPLC (Waters). The samples were pumped by 0.1% FA solution at 100 µL/min. The pepsin-digested peptides were then resolved by reverse-phased liquid chromatography using a C18 trap (Vanguard, Waters) and a C18 column (ACQUITYTM, Waters) maintained at 0–3°C (*Wales et al., 2008*). A gradient of 8–40% solvent A (0.1% FA in LCMS-grade water) to solvent B (0.1% FA in ACN) pumped at 40 µL/min by M-class binary solvent manager (Waters) was used to separate the peptides and identified through a coupled high-resolution Synapt G2-Si (Waters, UK) mass spectrometer. The peptides are ionized by electrospray ionization and detected in positive polarity mode. Ionized peptides were separated using HDMS$^E$ mode with ion mobility activated with the following parameters: low collision energy – trap (2 V), transfer (4 V); high collision energy – trap (4 V), transfer ramp (15–40 V); IMS wave velocity – 600 m/s, transfer wave velocity – 197 m/s; step-wave height at 30 V. A capillary voltage of 3 kV and 80°C source temperature were applied for ionization to detect the peptides and measure their absolute masses. For mass accuracy, 10 µL/min of 200 fmol/µL of Glu-fibrinogen peptide B was simultaneously sprayed as lockmass reference.

The mass spectrometry raw data acquired was processed using Protein Lynx Global Server v3.0 (Waters) to identify and annotate the peptides. Non-deuterated samples were used to match against the respective amino acid sequences of NS1c and Fab562 as the search database. The following search parameters were used for accurate matching and annotation: non-specific protease digestion, number of matches within peptide – 3, within protein – 7, a false positive rate of 4%, and a mass error of 1 ppm for precursor peptide. The annotated peptide list obtained was then filtered and analyzed using DynamX 3.0 (Waters) software. Peptides with a minimum intensity of 1000, minimum products per amino acid 0.1, and an MH$^+$ error of 10 ppm were used to filter out the poor quality and low-scoring peptides. This peptide list obtained from the non-deuterated controls was then used for deuterium exchange analysis using DynamX 3.0. The deuterium exchange data analyzed for the peptides in different sample conditions and across all time points was manually verified. Deuterium uptake was calculated as the difference in the masses of the centroids of the deuterated state and the corresponding non-deuterated control. The differences in deuterium exchange values between two states of the proteins were then checked for statistical significance using Deuteros 2.0 (*Lau et al., 2021*). All peptides with p-value<0.05 were only considered significant. The final analysis yielded 54 peptides covering 95% of NS1c, 31 peptides covering 94% of heavy chain of Fab562, and 33 peptides covering 92.5% of the light chain of Fab562. All HDX experiments were carried out in triplicate independent measurements, and the average deuterium exchange data is tabulated in *Supplementary file 6*. Raw files are accessible via JPOST.

## Acknowledgements

We thank the scientific facility support from the NTU Institute of Structural Biology and Protein Product Platform. The authors acknowledge the Cryo-Electron Microscopy Facility at Center for Bioimaging Science, Department of Biological Science, National University of Singapore, for scientific and technical assistance. Recombinant DENV2 NS1 protein is a generous gift from Dr. Shu Bo

and Professor Lok Shee Mei. We thank members of the DL and SV labs for their support. This research was supported by the Singapore Ministry of Education under its Singapore Ministry of Education Academic Research Fund Tier 2 (T2EP30220-0020) to DL and National Medical Research Council of Singapore (MOH-OFIRG18may-0006; MOH-OFIRG20nov-0002) to SV, A*STAR core funding and Singapore National Research Foundation under its NRF-SIS 'SingMass' scheme to RMS, Career Development Fund 2021 (A*STAR BMRC) to WWP and SZ, and LKCMedicine Dean's Postdoctoral Fellowship to CBLA.

## Additional information

### Funding

| Funder | Grant reference number | Author |
|---|---|---|
| Ministry of Education - Singapore | T2EP30220-0020 | Dahai Luo |
| National Medical Research Council | MOH-OFIRG20nov-0002 | Subhash G Vasudevan |
| National Medical Research Council | | Radoslaw M Sobota |
| Agency for Science, Technology and Research | | Wint Wint Phoo Zheng Ser Radoslaw M Sobota |
| Lee Kong Chian School of Medicine, Nanyang Technological University | LKCMedicine Dean's Postdoctoral Fellowship | Bing Liang Alvin Chew |
| National Medical Research Council | MOHOFIRG18may-0006 | Subhash G Vasudevan |
| Agency for Science, Technology and Research | Career Development Fund 2021 | Wint Wint Phoo Subhash G Vasudevan |

The funders had no role in study design, data collection and interpretation, or the decision to submit the work for publication.

### Author contributions

Bing Liang Alvin Chew, AN Qi Ngoh, Wint Wint Phoo, Zheng Ser, Data curation, Investigation, Methodology, Writing – original draft, Writing – review and editing; Kitti Wing Ki Chan, Data curation, Methodology, Writing – original draft, Project administration, Writing – review and editing; Nikhil K Tulsian, Data curation, Methodology, Writing – original draft; Shiao See Lim, Mei Jie Grace Weng, Satoru Watanabe, Milly M Choy, Data curation; Jenny Low, Eng Eong Ooi, Data curation, Formal analysis, Writing – review and editing; Christiane Ruedl, Data curation, Formal analysis; Radoslaw M Sobota, Data curation, Formal analysis, Methodology; Subhash G Vasudevan, Dahai Luo, Conceptualization, Data curation, Supervision, Writing – original draft, Project administration, Writing – review and editing

### Author ORCIDs

Bing Liang Alvin Chew http://orcid.org/0000-0002-2748-9366
AN Qi Ngoh http://orcid.org/0009-0006-2059-7854
Eng Eong Ooi http://orcid.org/0000-0002-0520-1544
Christiane Ruedl https://orcid.org/0000-0002-5599-6541
Subhash G Vasudevan http://orcid.org/0000-0002-5083-3831
Dahai Luo http://orcid.org/0000-0001-7637-7275

### Ethics

All animal experiments (protocol 2021/SHS/1646) were approved by the Institutional Animal Care And Use Committee at Singapore Health Services and conformed to the National Institutes of Health (NIH) guidelines and public law.

Reviewer #1 (Public Review): https://doi.org/10.7554/eLife.90762.3.sa1
Reviewer #2 (Public Review): https://doi.org/10.7554/eLife.90762.3.sa2
Author response https://doi.org/10.7554/eLife.90762.3.sa3

## Additional files

### Supplementary files

• MDAR checklist

• Supplementary file 1. Raw file of label-free quantification (LFQ) analysis of purified isNS1 proteins using liquid chromatography–mass spectrometry (LC-MS).

• Supplementary file 2. Raw file of crosslinking mass spectrometry (XL-MS) analysis of purified isNS1 proteins generated from metamorpheus search for both isNS1wt and isNS1ts.

• Supplementary file 3. Raw file of crosslinking mass spectrometry (XL-MS) crosslink distances between C-alphas when the identfied links are mapped on the predicted and fitted protein models of purified isNS1.

• Supplementary file 4. Raw file of crosslinking mass spectrometry (XL-MS) analysis of purified isNS1 proteins generated from metamorpheus search for DENV2 rsNS1-ApoA1.

• Supplementary file 5. Raw file of crosslinking mass spectrometry (XL-MS) crosslink distances between C-alphas when the identfied links are mapped on the different NS1 oligomer models.

• Supplementary file 6. Raw file of hydrogen-deuterium exchange mass spectrometry (HDX-MS) analysis for different peptides of NS1c in the presence and absence of Fab562 at various labeling times.

### Data availability

The data that support this study are available in the manuscript and supporting files. The cryoEM maps have been deposited in the Electron Microscopy Data Bank (EMDB) under accession codes EMD-36483 (isNS1ts:Fab56.2) and EMD-36480 (Fab56.2:isNS1ts:HDL). The corresponding PDB models for isNS1ts:Fab56.2 and Fab56.2:isNS1ts:HDL are available as source files because it is generated based on predicted models and a rigid body fit into the cryoEM map constraints. Crosslinking MS raw files and the search results can be downloaded from https://repository.jpostdb.org/entry/JPST001925. The HDX-MS data is deposited to the ProteomeXchange consortium via PRIDE partner repository with the dataset identifier PXD042235.

The following datasets were generated:

| Author(s) | Year | Dataset title | Dataset URL | Database and Identifier |
|---|---|---|---|---|
| Chew BLA, Luo D | 2023 | CryoEM structure of isNS1 in complex with Fab56.2 | https://www.emdataresource.org/EMD-36483 | EMDataResource, EMD-36483 |
| Chew BLA, Luo D | 2023 | CryoEM structure of isNS1 in complex with Fab56.2 and HDL | https://www.emdataresource.org/EMD-36480 | EMDataResource, EMD-36480 |
| Sobota R, Zheng S, Phoo WW, Luo D | 2023 | Secreted dengue virus NS1 is predominantly dimeric and in complex with high-density lipoprotein | https://repository.jpostdb.org/entry/JPST001925 | Jpost repository, JPST001925 |
| Tulsian NK, Luo D | 2024 | Secreted dengue virus NS1 is predominantly dimeric and incomplex with high-density lipoprotein | https://www.ebi.ac.uk/pride/archive/projects/PXD042235 | PRIDE, PXD042235 |

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
