## [Editor Report · eLife assessment]

This potentially **useful** study aims to advance our understanding of the structure of the native form of a viral toxin secreted from infected cells. While some of the findings confirm previous reports, the new claims in this study are unfortunately only **inadequately** supported by the methods and analyses used. More rigorous approaches are needed to justify the main conclusion that the structure of the viral toxin derived from infected cells in this study is distinct from previously reported structures of recombinantly expressed versions of the toxin.

---

## [Referee Report · Reviewer #1 (Public Review)]

The authors of this study seek to visualize NS1 purified from dengue virus infected cells. They infect vero cells with DV2-WT and DV2 NS1-T164S (a mutant virus previously characterized by the authors). The authors utilize an anti-NS1 antibody to immunoprecipitate NS1 from cell supernatants and then elute the antibody/NS1 complex with acid. The authors evaluate the eluted NS1 by SDS-PAGE, Native Page, mass spec, negative-stain EM, and eventually Cryo-EM. SDS-PAGE, mas spec, and native page reveal a >250 Kd species containing both NS1 and the proteinaceous component of HDL (ApoA1). The authors produce evidence to suggest that this population is predominantly NS1 in complex with ApoA1. This contrasts with recombinantly produced NS1 (obtained from a collaborator) which did not appear to be in complex with or contain ApoA1 (Figure 1C). The authors then visualize their NS1 stock in complex with their monoclonal antibody by CryoEM. For NS1-WT, the major species visualized by the authors was a ternary complex of an HDL particle in complex with an NS1 dimer bound to their mAB. For their mutant NS1-T164S, they find similar structures, but in contrast to NS1-WT, they visualize free NS1 dimers in complex with 2 Fabs (similar to what's been reported previously) as one of the major species. This highlights that different NS1 species have markedly divergent structural dynamics. It's important to note that the electron density maps for their structures do appear to be a bit overfitted since there are many regions with electron density that do not have a predicted fit and their HDL structure does not appear to have any predicted secondary structure for ApoA1. The authors then map the interaction between NS1 and ApoA1 using cross-linking mass spectrometry revealing numerous NS1-ApoA1 contact sites in the beta-roll and wing domain. The authors find that NS1 isolated from DENV infected mice is also present as a >250 kD species containing ApoA1. They further determine that immunoprecipitation of ApoA1 out of the sera from a single dengue patient correlates with levels of NS1 (presumably COIPed by ApoA1) in a dose-dependent manner.

In the end, the authors make some useful observations for the NS1 field (mostly confirmatory) providing additional insight into the propensity of NS1 to interact with HDL and ApoA1. The study does not provide any functional assays to demonstrate activity of their proteins or conduct mutagenesis (or any other assays) to support their interaction predications. The authors assertion that higher-order NS1 exists primarily as a NS1 dimer in complex with HDL is not well supported as their purification methodology of NS1 likely introduces bias as to what NS1 complexes are isolated. While their results clearly reveal NS1 in complex with ApoA1, the lack of other NS1 homo-oligomers may be explained by how they purify NS1 from virally infected supernatant. Because NS1 produced during viral infection is not tagged, the authors use an anti-NS1 monoclonal antibody to purify NS1. This introduces a source of bias since only NS1 oligomers with their mAb epitope exposed will be purified. Further, the use of acid to elute NS1 may denature or alter NS1 structure and the authors do not include controls to test functionality of their NS1 stocks (capacity to trigger endothelial dysfunction or immune cell activation). The acid elution may force NS1 homo-oligomers into dimers which then reassociate with ApoA1 in a manner that is not reflective of native conditions. Conducting CryoEM of NS1 stocks only in the presence of full-length mAbs or Fabs also severely biases what species of NS1 is visualized since any NS1 oligomers without the B-ladder domain exposed will not be visualized. If the residues obscured by their mAb are involved in formation of higher-order oligomers then this antibody would functionally inhibit these species from forming. The absence of critical controls, use of one mAb, and acid elution for protein purification severely limits the interpretation of these data and do not paint a clear picture of if NS1 produced during infection is structurally distinct from recombinant NS1. Certainly there is novelty in purifying NS1 from virally infected cells, but without using a few different NS1 antibodies to purify NS1 stocks (or better yet a polyclonal population of antibodies) it's unclear if the results of the authors are simply a consequence of the mAb they selected.

Data produced from numerous labs studying structure and function of flavivirus NS1 proteins provide diverse lines of evidence that the oligomeric state of NS1 is dynamic and can shift depending on context and environment. This means that the methodology used for NS1 production and purification will strongly impact the results of a study. The data in this manuscript certainly capture one of these dynamic states and overall support the general model of a dynamic NS1 oligomer that can associate with both host proteins as well as itself but the assertions of this manuscript are overall too strong given their data, as there is little evidence in this manuscript, and none available in the large body of existing literature, to support that NS1 exists only as a dimer associated with ApoA1. More likely the results of this paper are a result of their NS1 purification methodology.

Comments on revised version:

The authors have not adequately addressed my concerns from the original review. My major concerns are that the binding modality of NS1 to ApoA1/HDL was not validated using a mutagenesis approach and that the overarching conclusion drawn by the authors, that the major species of NS1 in vivo is a dimer in complex with ApoA1, is not supported by the data in this study given the methodology of using a single monoclonal antibody to immunoprecipitate NS1. Certainly, the structures in this manuscript are valuable in confirming that NS1 interacts with HDL and captures a snapshot of NS1/HDL interaction dynamics, but the use of only a single antibody is a major source of bias that makes it challenging to draw conclusions about the oligomeric state of NS1. Further on this point, a critically important control that is missing from this study is to determine if the anti-NS1 mAb 56.2 prevents NS1 from interacting with cells, triggering the release of proinflammatory cytokines from immune cells, or mediating endothelial dysfunction of endothelial cells. If this antibody inhibits these NS1-triggered events (linked to pathogenesis), it would suggest that the NS1 within this ternary complex is not active. Presumably some protective anti-NS1 antibodies may function by modulating the oligomeric state of NS1.

---

## [Referee Report · Reviewer #2 (Public Review)]

Summary:

Chew et al describe interaction of the flavivirus protein NS1 with HDL using primarily cryoEM and mass spec. The NS1 was secreted from dengue virus infected Vero cells, and the HDL were derived from the 3% FBS in the culture media. NS1 is a virulence factor/toxin and is a biomarker for dengue infection in patients. The mechanisms of its various activities in the host are incompletely understood. NS1 has been seen in dimer, tetramer and hexamer forms. It is well established to interact with membrane surfaces, presumably through a hydrophobic surface of the dimer form, and the recombinant protein has been shown to bind HDL. In this study, cryoEM and crosslinking-mass spec are used to examine NS1 secreted from virus-infected cells, with the conclusion that the sNS1 is predominantly/exclusively HDL-associated through specific contacts with the ApoA1 protein.

Strengths: The experimental results are consistent with previously published data.

Weaknesses

CryoEM:

Some of the neg-stain 2D class averages for sNS1 in Fig S1 clearly show 1 or 2 NS1 dimers on the surface of a spherical object, presumably HDL, and indicate the possibility of high-quality cryoEM results. However, the cryoEM results are disappointing. The cryo 2D class averages and refined EM map in Fig S4 are of poor quality, indicating sub-optimal grid preparation or some other sample problem. Some of the FSC curves (2 in Fig S7 and 1 in Fig S6) have extremely peculiar shapes, suggesting something amiss in the map refinement. The sharp drop in the "corrected" FSC curves in Figs S5c and S6c (upper) indicate severe problems. The stated resolutions (3.42 & 3.82 Å) for the sNS1ts-Fab56.2 are wildly incompatible with the images of the refined maps in Figs 3 & S7. At those resolutions, clear secondary structural elements should be visible throughout the map. From the 2D averages and 3D maps shown in the figures, this does not seem to be the case. Local resolution maps should be shown for each structure.

The samples were clearly challenging for cryoEM, leading to poor quality maps that were difficult to interpret. None of the figures are convincing that NS1, Ab56.2 or Fab56.2 are correctly fit into EM maps. There is no indication of ApoA1 helices. Details of the fit of models to density for key regions of the higher-resolution EM maps should be shown and the models should be deposited in the PDB. An example of modeling difficulty is clear in the sNS1ts dimer with bound Fab56.2 (figs 3c & S7e). For this complex, the orientation of the Fab56.2 relative to the sNS1ts dimer in this submission (Fig 3c) is substantially different than in the bioRxiv preprint (Fig 3c). Regions of empty density in Fig 3c also illustrate the challenge of building a model into this map.

Mass spec:

Crosslinking-mass spec was used to detect contacts between NS1 and ApoA1, providing strong validation of the sNS1-HDL association. As the crosslinks were detected in a bulk sample, they show that NS1 is near ApoA1 in many/most HDL particles, but they do not indicate a specific protein-protein complex. Thus, the data do not support the model of an NS1-ApoA1 complex in Fig 4d. Further, a specific NS1-ApoA1 interaction should have evidence in the EM maps (helical density for ApoA1), but none is shown or mentioned. If such exists, it could perhaps be visualized after focused refinement of the map for sNS1ts-HDL with Fab56.2 (Fig S7d). The finding that sNS1-ApoA1 crosslinks involved residues on the hydrophobic surface of the NS1 dimer confirms previous data that this NS1 surface engages with membranes and lipids.

Sample quality:

The paper lacks any validation that the purified sNS1 retains established functions, for example the ability to enhance virus infectivity or to promote endothelial dysfunction. Peculiarities include the gel filtration profiles (Fig 2a), which indicate identical elution volumes (apparent MWs) for sNS1wt-HDL bound to Ab562 (~150 kDa) and to the ~3X smaller Fab56.2 (~50 kDa). There should also be some indication of sNS1wt-HDL pairs crosslinked by the full-length Ab, as can be seen in the raw cryoEM micrograph (Fig S5b).

Obtaining high quality structures is often more demanding of sample integrity than are activity assays. Given the low quality of the cryoEM maps, it's possible that the acidification step in immunoaffinity purification damaged the HDL complex. No validation of HDL integrity, for example with acid-treated HDL, is reported. Acid treatment is perhaps discounted by a statement (line 464) that another group also used immunoaffinity purification in a recent study (ref 20) reporting sNS1 bound to HDL. However the statement is incorrect; the cited study used affinity purification via a strep-tag on recombinant sNS1.

Discussion:

The Discussion reflects a view that the NS1 secreted from virus-infected cells is a 1:1 sNS1dimer:HDL complex with the specific NS1-ApoA1 contacts detected by crosslinking mass spec. This is inconsistent with both the neg-stain 2D class average with 2 sNS1 dimers on an HDL (Fig S1c) and with the recent study of Flamand & co-workers showing 1-3 NS1 dimers per HDL (ref 20). It also ignores the propensity of NS1 to associate with membranes and lipids. It is far more likely that NS1 association with HDL is driven by these hydrophobic interactions than by specific protein-protein contacts. A lengthy Discussion section (lines 461-522) includes several chemically dubious or inconsistent statements, all based on the assumption that specific ApoA1 contacts are essential to NS1 association with HDL and that sNS1 oligomers higher than the dimer necessarily involve ApoA1 interaction, conclusions that are not established by the data in this paper.

Additional comments on the revised manuscript:

Comments on the structures:

The authors kindly provided their fitted atomic models for the 2 reported structures. The EM maps are available in the EMDB. Based on these materials, the derived structures are not well supported due to problems with the models, the maps, and the fit of models to maps.

Quick inspection revealed that the models for both structures are implausible due to a large steric clash of Fab56.2 and the end of the NS1. The Fab and NS1 protein backbones interpenetrate by nearly 20 Å. This substantial overlap exists for all 3 Fab56.2-NS1 interactions in the 2 structures, and is also visible in the perpendicular views of the NS1 dimer with 2 bound Fab56.2 in Fig. 2c. It appears that the Fab56.2 model was jammed into the NS1 model in order to bring all domains inside the density envelope at the threshold chosen to display the map. The poor fit of model to map is also clear in several protruding density regions without any model.

The fits of both atomic models to the maps are questionable because

- The maps suffer from severe preferred orientation problems, as seen in the streaky tubes of density. The streaks in both maps do not match the NS1 beta strands of the fitted models.

- The shape of the modeled ApoA1 helical ring surrounding the HDL does not match the shape of the EM density. In some regions, the ApoA1 helices are inside the rather strong density for the spherical HDL, but in other regions the helices are outside the density.

- Both maps have regions of strong density that are adjacent to NS1 but lack any protein model, while other parts of the structure, including the beta-roll domain, lack density.

- The claimed 4.3-Å resolution of the NS1-Fab56.2 complex is wildly overstated. The local resolution of ~2.5 Å for the "best" part of the structure (Supp Fig. 7E) appears to pertain to the beta strands at the center of the NS1 dimer. However, these density streaks do not match the beta strands of the fit model.

- The manuscript lacks statistics on the fit of model to map. A standard cryo-EM "Table 1" should include more than is presented in Supp Table 1. The fitted model for at least the higher resolution structure should be deposited in the PDB.

Comments on the structure interpretation:

By now it should be abundantly clear that the oligomer state of NS1 is dynamic and highly sensitive to environmental conditions and to each sample's "history". For the reasons pointed out by reviewer 1, it is not clear that the immunoaffinity purification method captured all forms of sNS1 equally. Thus, the authors insistence that NS1 secreted from virus-infected cells is predominantly bound to HDL particles in a ratio of 1 NS1 dimer per HDL is not well supported. They employ similar arguments to challenge the discovery of sNS1 as a lipoprotein particle (PNAS 2011), contending that the 2011 finding was an artefact of recombinant NS1 production and is irrelevant to sNS1 from a virus infection. The several published structures of NS1 oligomers reveal a large degree of asymmetry in dimer-dimer interaction, consistent with the ability of NS1 to dynamically associate with a variety of hydrophobic entities.

---

## [Author Response]

The following is the authors’ response to the original reviews.

**Public Reviews:**

**Reviewer #1 (Public Review):**
The authors of this study seek to visualize NS1 purified from dengue virus infected cells. They infect vero cells with DV2-WT and DV2 NS1-T164S (a mutant virus previously characterized by the authors). The authors utilize an anti-NS1 antibody to immunoprecipitate NS1 from cell supernatants and then elute the antibody/NS1 complex with acid. The authors evaluate the eluted NS1 by SDS-PAGE, Native Page, mass spec, negative-stain EM, and eventually Cryo-EM. SDS-PAGE, mas spec, and native page reveal a >250 Kd species containing both NS1 and the proteinaceous component of HDL (ApoA1). The authors produce evidence to suggest that this population is predominantly NS1 in complex with ApoA1. This contrasts with recombinantly produced NS1 (obtained from a collaborator) which did not appear to be in complex with or contain ApoA1 (Figure 1C). The authors then visualize their NS1 stock in complex with their monoclonal antibody by CryoEM. For NS1-WT, the major species visualized by the authors was a ternary complex of an HDL particle in complex with an NS1 dimer bound to their mAB. For their mutant NS1-T164S, they find similar structures, but in contrast to NS1-WT, they visualize free NS1 dimers in complex with 2 Fabs (similar to what's been reported previously) as one of the major species. This highlights that different NS1 species have markedly divergent structural dynamics. It's important to note that the electron density maps for their structures do appear to be a bit overfitted since there are many regions with electron density that do not have a predicted fit and their HDL structure does not appear to have any predicted secondary structure for ApoA1. The authors then map the interaction between NS1 and ApoA1 using cross-linking mass spectrometry revealing numerous NS1-ApoA1 contact sites in the beta-roll and wing domain. The authors find that NS1 isolated from DENV infected mice is also present as a >250 kD species containing ApoA1. They further determine that immunoprecipitation of ApoA1 out of the sera from a single dengue patient correlates with levels of NS1 (presumably COIPed by ApoA1) in a dose-dependent manner.In the end, the authors make some useful observations for the NS1 field (mostly confirmatory) providing additional insight into the propensity of NS1 to interact with HDL and ApoA1. The study does not provide any functional assays to demonstrate activity of their proteins or conduct mutagenesis (or any other assays) to support their interaction predications. The authors assertion that higher-order NS1 exists primarily as a NS1 dimer in complex with HDL is not well supported as their purification methodology of NS1 likely introduces bias as to what NS1 complexes are isolated. While their results clearly reveal NS1 in complex with ApoA1, the lack of other NS1 homo-oligomers may be explained by how they purify NS1 from virally infected supernatant. Because NS1 produced during viral infection is not tagged, the authors use an anti-NS1 monoclonal antibody to purify NS1. This introduces a source of bias since only NS1 oligomers with their mAb epitope exposed will be purified. Further, the use of acid to elute NS1 may denature or alter NS1 structure and the authors do not include controls to test functionality of their NS1 stocks (capacity to trigger endothelial dysfunction or immune cell activation). The acid elution may force NS1 homo-oligomers into dimers which then reassociate with ApoA1 in a manner that is not reflective of native conditions. Conducting CryoEM of NS1 stocks only in the presence of full-length mAbs or Fabs also severely biases what species of NS1 is visualized since any NS1 oligomers without the B-ladder domain exposed will not be visualized. If the residues obscured by their mAb are involved in formation of higher-order oligomers then this antibody would functionally inhibit these species from forming. The absence of critical controls, use of one mAb, and acid elution for protein purification severely limits the interpretation of these data and do not paint a clear picture of if NS1 produced during infection is structurally distinct from recombinant NS1. Certainly there is novelty in purifying NS1 from virally infected cells, but without using a few different NS1 antibodies to purify NS1 stocks (or better yet a polyclonal population of antibodies) it's unclear if the results of the authors are simply a consequence of the mAb they selected.Data produced from numerous labs studying structure and function of flavivirus NS1 proteins provide diverse lines of evidence that the oligomeric state of NS1 is dynamic and can shift depending on context and environment. This means that the methodology used for NS1 production and purification will strongly impact the results of a study. The data in this manuscript certainly capture one of these dynamic states and overall support the general model of a dynamic NS1 oligomer that can associate with both host proteins as well as itself but the assertions of this manuscript are overall too strong given their data, as there is little evidence in this manuscript, and none available in the large body of existing literature, to support that NS1 exists only as a dimer associated with ApoA1. More likely the results of this paper are a result of their NS1 purification methodology.Suggestions for the Authors:Major:(1) Because of the methodology used for NS1 purification, it is not clear from the data provided if NS1 from viral infection differs from recombinant NS1. Isolating NS1 from viral infection using a polyclonal antibody population would be better to answer their questions. On this point, Vero cells are also not the best candidate for their NS1 production given these cells do not come from a human. A more relevant cell line like U937-DC-SIGN would be preferable.

We performed an optimization of sNS1 secretion from DENV infection in different cell lines (Author response image 1 below) to identify the best cell line candidate to obtain relatively high yield of sNS1 for the study. As shown in Author response image 1, the levels of sNS1 in the tested human cell lines Huh7 and HEK 293T were at least 3-5 fold lower than in Vero cells. Although using a monocytic cell line expressing DC-SIGN as suggested by the reviewer would be ideal, in our experience the low infectivity of DENV in monocytic cell lines will not yield sufficient amount of sNS1 needed for structural analysis. For these practical reasons we decided to use the closely related non-human primate cell line Vero for sNS1 production supported by our optimization data.

**Author response image 1. sa3fig1:** sNS1 secretion in different mammalian and mosquito cell lines after DENV2 infection. The NS1 secretion level is measured using PlateliaTM Dengue NS1 Ag ELISA kit (Bio-Rad) on day 3 (left) and day 5 (right) post infection respectively.

(2) The authors need to support their interaction predictions and models via orthogonal assays like mutagenesis followed by HDL/ApoA1 complexing and even NS1 functional assays. The authors should be able to mutate NS1 at regions predicted to be critical for ApoA1/HDL interaction. This is critical to support the central conclusions of this manuscript.

In our previous publication (Chan et al., 2019 Sci Transl Med), we used similarly purified sNS1 (immunoaffinity purification followed by acid elution) from infected culture supernatants from both DENV2 wild-type and T164S mutant (both also studied in the present work) to carry out stimulation assay on human PBMCs as described by other leading laboratories investigating NS1 (Modhiran et al., 2015 Sci Transl Med). For reader convenience we have extracted the data from our published paper and present it as Author response image 2 below.

**Author response image 2. sa3fig2:** (A) IL6 and (B) TNFa concentrations measured in the supernatants of human PBMCs incubated with either 1µg/ml or 10µg/ml of the BHK-21 immunoaffinity-purified WT and TS mutant sNS1 for 24 hours. Data is adapted from Chan et al., 2019.

Incubation of immunoaffinity-purified sNS1 (WT and TS) with human PBMCs from 3 independent human donors triggered the production of proinflammatory cytokines IL6 and TNFα in a concentration dependent manner (Author response image 2), consistent with the published data by Modhiran et al., 2015 Sci Transl Med. Interestingly the TS mutant derived sNS1 induced a higher proinflammatory cytokines production than WT virus derived sNS1 that appears to correlate with the more lethal and severe disease phenotype in mice as also reported in our previous work (Chan et al., 2019). Additionally, the functionality of our immune-affinity purified infection derived sNS1 (isNA1) is now further supported by our preliminary results on the NS1 induced endothelial cell permeability assay using the purified WT and mutant isNS1 (Author response image 3). As shown in Author response image 3, both the isNS1wt and isNS1ts mutant reduced the relative transendothelial resistance from 0 to 9 h post-treatment, with the peak resistance reduction observed at 6 h post-treatment, suggesting that the purified isNS1 induced endothelial dysfunction as reported in Puerta-Guardo et al., 2019, Cell Rep. It is noteworthy that the isNS1 in our study behaves similarly as the commercial recombinant sNS1 (rsNS1 purchased from the same source used in study by Puerta-Guardo et al., 2019) in inducing endothelial hyperpermeability. Collectively our previous published and current data suggest that the purified isNS1 (as a complex with ApoA1) has a pathogenic role in disease pathogenesis that is also supported in a recent publication by Benfrid et al., EMBO 2022. The acid elution has not affected the functionality of NS1.

**Author response image 3. sa3fig3:** Functional assessment of isNS1wt and isNS1ts on vascular permeability in vitro. A trans-endothelial permeabilty assay via measurement of the transendothelial electrical resistance (TEER) on human umbilical vascular endothelial cells (hUVEC) was performed, as described previously (Puerta-Guardo et al., 2019, Cell Rep). Ovalbumin serves as the negative control, while TNF-α and rsNS1 serves as the positive controls.

We agree with reviewer about the suggested mutagnesis study. We will perform site-directed mutagenesis at selected residues and further structural and functional analyses and report the results in a follow-up study.

(3) The authors need to show that the NS1 stocks produced using acid elution are functional compared to standard recombinantly produced NS1. Do acidic conditions impact structure/function of NS1?

We are providing the same response to comments 1 & 2 above. We would like to reiterate that we have previously used sNS1 from immunoaffinity purification followed by acid elution to test its function in stimulating PBMCs to produce pro-inflammatory cytokines (Chan et al., 2019; Author response image 2). Similar to Modhiran et al. (2015) and Benfrid et al. (2022), the sNS1 that we extracted using acid elution are capable of activating PBMCs to produce pro-inflammatory cytokines. We have now further demonstrated the ability of both WT and TS isNS1 in inducing endothelial permeability in vitro in hUVECs, using the TEER assay (Author response image 3). Based on the data presented in the rebuttal figures as well as our previous publication we do not think that the acid elution has a significant impact on function of isNS1.

We performed affinity purification to enrich the complex for better imaging and analysis (Supp Fig. 1b) since the crude supernatant contains serum proteins and serum-free infections also do not provide sufficient isNS1. The major complex observed in negative stain is 1:1 (also under acidic conditions which implies that the complex are stable and intact). We agree that it is possible that other oligomers can form but we have observed only a small population (74 out of 3433 particles, 2.15%; 24 micrographs) of HDL:sNS1 complex at 1:2 ratio as shown in the Author response image 4 below and in the manuscript (p. 4 lines 114-117, Supp Fig. 1c). Other NS1 dimer:HDL ratios including 2:1 and 3:1 have been reported by Benfrid et al., 2022 by spiking healthy sera with recombinant sNS1 and subsequent re-affinity purification. However, this method used an approximately 8-fold higher sNS1 concentration (400 ug/mL) than the maximum clinically reported concentration (50 ug/mL) (Young et al., 2000; Alcon et al., 2002; Libraty et al., 2002). In our hands, the sNS1 concentration in the concentrated media from in vitro infection was quantified as 30 ug/mL which is more physiologically relevant.

We conclude that the integrity of the HDL of the complex is not lost during sample preparation, as we are able to observe the complex under the negative staining EM as well as infer from XL-MS. Our rebuttal data and our previous studies with our acid-eluted isNS1 from immunoaffinity purification clearly show that our protein is functional and biologically relevant.

**Author response image 4. sa3fig4:** (**A**) Representative negative stain micrograph of sNS1wt (**B**) Representative 2D averages of negative stained isNS1wt. Red arrows indicating the characteristic wing-like protrusions of NS1 inserted in HDL. (**C**) Data adapted from Figure 2 in Benfrid et al. (2022).

(4) Overall, the data obtained from the mutant NS1 (contrasted to WT NS1) reveals how dynamic the oligomeric state of NS1 proteins are but the authors do not provide any insight into how/why this is, some additional lines of evidence using either structural studies or mutagenesis to compare WT and their mutant and even NS1 from a different serotype of DENV would help the field to understand the dynamic nature of NS1.

The T164S mutation in DENV2 NS1 was proposed as the residue associated with disease severity in 1997 Cuban dengue epidemic Halsted SB. “Intraepidemic increases in dengue disease severity: applying lessons on surveillance and transmission”. Whitehorn, J., Farrar. J., Eds., Clinical Insights in Dengue: Transmission, Diagnosis & Surveillance. The Future Medicine (2014), pp. 83-101. Our previous manuscript examined this mutation by engineering it into a less virulent clade 2 DENV isolated in Singapore and showed that sNS1 production was higher without any change in viral RNA replication. Transcript profiling of mutant compared to WT virus showed that genes that are usually induced during vascular leakage were upregulated for the mutant. We also showed that infection of interferon deficient AG129 mice with the mutant virus resulted in disease severity, increased complement protein expression in the liver, tissue inflammation and greater mortality compared to WT virus infected mice. The lipid profiling in our study (Chan et al., 2019) suggested small differences with WT but was overall similar to HDL as described by Gutsche et al. (2011). We were intrigued by our functional results and wanted to explore more deeply the impact of the mutation on sNS1 structure which at that stage was widely believed to be a trimer of NS1 dimers with a central channel (~ X Å) stuffed with lipid as established in several seminal publications (Flamand et al., 1999; Gutsche et al., 2011; Muller et al., 2012). In fact “This Week in Virology” netcast (https://www.microbe.tv/twiv/twiv-725/) discussed two back-to-back publications in Science (Modhiran et al., 371(6625)190-194; Biering et al., Science 371(6625):194-200) which showed that therapeutic antibodies can ameliorate the NS1 induced pathogenesis and expert discussants posed questions that also pointed to the need for more accurate definition of the molecular composition and architecture of the circulating NS1 complex during virus infection to get a clearer handle on its pathogenic mechanism. Our current studies and also the recent high resolution cryoEM structures (Shu et al., 2022) do not support the notion of a central channel “stuffed with lipid”. Even in the rare instances where trimer of dimers are shown, the narrow channel in the center could only accommodate one molecule of lipoid molecule no bigger than a typical triglyceride molecule. This hexamer model cannot explain the lipid proeotmics data in the literature.

In our study we observed predominantly 1:1 NS1 dimer to HDL (~30 μg/mL) mirroring maximum clinically reported concentration of sNS1 in the sera of DENV patients (40-50 μg/mL) as we highlighted in our main text (P. 18, lines 461-471). What is often quoted (also see later) is the recent study of Flamand & co-workers which show 1-3 NS1 dimers per HDL (Benfrid et al, 2022) by spiking rsNS1 (400 μg/mL) with HDL. This should not be confused with the previous models which suggested a lipid filled central channel holding together the hexamer. The use of physiologically relevant concentrations is important for these studies as we have highlighted in our main text (P. 18, lines 461-471).

Our interpretation for the mutant (isNS1ts) is that it is possible that the hydrophilic serine at residue 164 located in the greasy finger loop may weaken the isNS1ts binding to HDL hence the observation of free sNS1 dimers in our immunoaffinity purified (acid eluted sample). The disease severity and increased complement protein expression in AG129 mice liver can be ascribed to weakly bound mutant NS1 with fast on/off rate with HDL being transported to the liver where specific receptors bind to free sNS1 and interact with effector proteins such as complement to drive inflammation and associated pathology. Our indirect support for this is that the XL-MS analysis of purified isNS1ts identified only 7 isNS1ts:ApoA1 crosslinks while 25 isNS1wt:ApoA1 crosslinks were identified from purified isNS1wt (refer to Fig. 4 and Supp. Fig. 8).

Taken together, the cryoEM and XL-MS analysis of purified isNS1ts suggest that isNS1ts has weaker affinity for HDL compared to isNS1wt. We welcome constructive discussion on our interpretation that we and others will hopefully obtain more data to support or deny our proposed explanation. Our focus has been to compare WT with mutant sNS1 from DENV2 and we agree that it will be useful to study other serotypes.

**Reviewer #2:**
CryoEM:Some of the neg-stain 2D class averages for sNS1 in Fig S1 clearly show 1 or 2 NS1 dimers on the surface of a spherical object, presumably HDL, and indicate the possibility of high-quality cryoEM results. However, the cryoEM results are disappointing. The cryo 2D class averages and refined EM map in Fig S4 are of poor quality, indicating sub-optimal grid preparation or some other sample problem. Some of the FSC curves (2 in Fig S7 and 1 in Fig S6) have extremely peculiar shapes, suggesting something amiss in the map refinement. The sharp drop in the "corrected" FSC curves in Figs S5c and S6c (upper) indicate severe problems. The stated resolutions (3.42 & 3.82 Å) for the sNS1ts-Fab56.2 are wildly incompatible with the images of the refined maps in Figs 3 & S7. At those resolutions, clear secondary structural elements should be visible throughout the map. From the 2D averages and 3D maps shown in the figures this does not seem to be the case. Local resolution maps should be shown for each structure.

The same sample is used for negative staining and the cryoEM results presented. The cryoEM 2D class averages are similar to the negative stain ones, with many spherical-like densities with no discernible features, presumably HDL only or the NS1 features are averaged out. The key difference lies in the 2D class averages where the NS1 could be seen. The side views of NS1 (wing-like protrusion) are more obvious in the negative stain while the top views of NS1 (cross shaped-like protrusion) are more obvious under cryoEM. HDL particles are inherently heterogeneous and known to range from 70-120 Å, this has been highlighted in the main text (p. 8, lines 203 and 228). This helps to explain why the reviewer may find the cryoEM result disappointing. The sample is inherently challenging to resolve structurally as it is (not that the sample is of poor quality). In terms of grid preparation, Supp Fig 4b shows a representative motion-corrected micrograph of the isNS1ts sample whereby individual particles can be discerned and evenly distributed across the grid at high density.

We acknowledge that most of the dips in the FSC curves (Fig S5-7) are irregular and affect the accuracy of the stated resolutions, particularly for the HDL-isNS1ts-Fab56.2 and isNS1ts-Fab56.2 maps for which the local resolution maps are shown (Fig S7d-e). Probable reasons affecting the FSC curves include (1) the heterogeneous nature of HDL, (2) preferred orientation issue (p 7, lines 198 -200), and (3) the data quality is intrinsically less ideal for high resolution single particle analysis. Optimizing of the dynamic masking such that the mask is not sharper than the resolution of the map for the near (default = 3 angstroms) and far (12 angstroms) parameters during data processing, ranging from 6 - 12 and 14 - 20 respectively, did not help to improve the FSC curves. To report a more accurate global resolution, we have revised the figures S5-7 with new FSC curve plots generated using the remote 3DFSC processing server.

Regardless, the overall architecture and the relative arrangement of NS1 dimer, Fab, and HDL are clearly visible and identifiable in the map. These results agree well with our biochemical data and mass-spec data.

The samples were clearly challenging for cryoEM, leading to poor quality maps that were difficult to interpret. None of the figures are convincing that NS1, Ab56.2 or Fab56.2 are correctly fit into EM maps. There is no indication of ApoA1 helices. Details of the fit of models to density for key regions of the higher-resolution EM maps should be shown and the models should be deposited in the PDB. An example of modeling difficulty is clear in the sNS1ts dimer with bound Fab56.2 (figs 3c & S7e). For this complex, the orientation of the Fab56.2 relative to the sNS1ts dimer in this submission (Fig 3c) is substantially different than in the bioRxiv preprint (Fig 3c). Regions of empty density in Fig 3c also illustrate the challenge of building a model into this map.

We acknowledge the modelling challenge posed by low resolution maps in general, such as the handedness of the Fab molecule as pointed out by the reviewer (which is why others have developed the use of anti-fab nanobody to aid in structure determination among other methods). The change in orientation of the Fab56.2 relative to the sNS1ts dimer was informed by the HDX-MS results which was not done at the point of bioRxiv preprint mentioned. With regards to indication of ApoA1 helices, this is expected given the heterogeneous nature of HDL. To the best of our knowledge, engineered apoA1 helices were also not reported in many cryoEM structures of membrane proteins solved in membrane scaffold protein (MSP) nanodiscs. This is despite nanodiscs, comprised of engineered apoA1 helices, having well-defined size classifications.

Regions of weak density in Fig 3c is expected due to the preferred orientation issue acknowledged in the results section of the main text (p. 9, line 245). The cryoEM density maps have been deposited in the Electron Microscopy Data Bank (EMDB) under accession codes EMD-36483 (isNS1ts:Fab56.2) and EMD-36480 (Fab56.2:isNS1ts:HDL). The protein model files for isNS1ts:Fab56.2 and Fab56.2:isNS1ts:HDL model are available upon request. Crosslinking MS raw files and the search results can be downloaded from JPOST. The HDX-MS data is deposited to the ProteomeXchange consortium via PRIDE partner repository (46) with the dataset identifier PXD042235.

Mass spec:Crosslinking-mass spec was used to detect contacts between NS1 and ApoA1, providing strong validation of the sNS1-HDL association. As the crosslinks were detected in a bulk sample, they show that NS1 is near ApoA1 in many/most HDL particles, but they do not indicate a specific protein-protein complex. Thus, the data do not support the model of an NS1-ApoA1 complex in Fig 4d. Further, a specific NS1-ApoA1 interaction should have evidence in the EM maps (helical density for ApoA1), but none is shown or mentioned. If such exists, it could perhaps be visualized after focused refinement of the map for sNS1ts-HDL with Fab56.2 (Fig S7d). The finding that sNS1-ApoA1 crosslinks involved residues on the hydrophobic surface of the NS1 dimer confirms previous data that this NS1 surface engages with membranes and lipids.

We thank the reviewer for the comment. The XL-MS is a method to identify the protein-protein interactions by proximity within the spacer arm length of the crosslinker. The crosslinking MS data do support the NS1-ApoA1 complex model obtained by cryo-EM because the identified crosslinks that are superimposed on the EM map are within the cut-off distance of 30 Å. We agree that the XL-MS data do not dictate the specific interactions between specific residues of NS1-ApoA1 in the EM model. We also do not claim that specific residue of NS1 in beta roll or wing domain is interacting with specific residue of ApoA1 in H4 and H5 domain. We claim that beta roll and wing domain regions of NS1 are interacting with ApoA1 in HDL indicating the proximity nature of NS1-ApoA1 interactions as warranted by the XL-MS data.

As explained in the previous response on the lack of indication of ApoA1 helical density, this is expected given the heterogeneous nature of HDL. It is typical to see lipid membranes as unstructured and of lower density than the structured protein. In our study, local refinement was performed on either the global map (presented in Fig S7d) or focused on the NS1-Fab region only. Both yielded similar maps as illustrated in the real space slices shown in Author response image 5. The mask and map overlay is depicted in similar orientations to the real space slices, and at different contour thresholds at 0.05 (Author response image 5) and 0.135 (Author response image 5). While the overall map is of poor resolution and directional anisotropy evident, there is clear signal differences in the low density region (i.e. the HDL sphere) indicative of NS1 interaction with ApoA1 in HDL, extending from the NS1 wing to the base of the HDL sphere.

**Author response image 5. sa3fig5:** Real Space Slices of map and mask used during Local Refinement for overall structure (**a-b**) and focused mask on NS1 region (**c-d**). The corresponding map (grey) contoured at 0.05 (**e**) and 0.135 (**f**) in similar orientations as shown for the real space slices of map and masks. The focused mask of NS1 used is colored in semi-transparent yellow. Real Space Slices of map and mask are generated during data processing in Cryosparc 4.0 and the map figures were prepared using ChimeraX.

Sample quality:The paper lacks any validation that the purified sNS1 retains established functions, for example the ability to enhance virus infectivity or to promote endothelial dysfunction.

Please see detailed response for question 2 in Reviewer #1’s comments. In essence, we have showed that both isNS1wt and isNS1ts are capable of inducing endothelial permeability in an in vitro TEER assay (Rebuttal Fig 3) and also in our previous study that quantified inflammation in human PBMC’s (Rebuttal Fig 2).

Peculiarities include the gel filtration profiles (Fig 2a), which indicate identical elution volumes (apparent MWs) for sNS1wt-HDL bound to Ab562 (~150 kDa) and to the ~3X smaller Fab56.2 (~50 kDa). There should also be some indication of sNS1wt-HDL pairs crosslinked by the full-length Ab, as can be seen in the raw cryoEM micrograph (Fig S5b).Obtaining high quality structures is often more demanding of sample integrity than are activity assays. Given the low quality of the cryoEM maps, it's possible that the acidification step in immunoaffinity purification damaged the HDL complex. No validation of HDL integrity, for example with acid-treated HDL, is reported.

Please see detailed response for question 3 in Reviewer #1’s comments.

Acid treatment is perhaps discounted by a statement (line 464) that another group also used immunoaffinity purification in a recent study (ref 20) reporting sNS1 bound to HDL. However the statement is incorrect; the cited study used affinity purification via a strep-tag on recombinant sNS1.

We thank the Reviewer for pointing this out and have rewritten this paragraph instead (p 18, line 445-455). We also expanded our discussion to highlight our prior functional studies showing that acid-eluted isNS1 proteins do induce endothelial hyperpermeability (p 18-19, line 470-476).

Discussion:The Discussion reflects a view that the NS1 secreted from virus-infected cells is a 1:1 sNS1dimer:HDL complex with the specific NS1-ApoA1 contacts detected by crosslinking mass spec. This is inconsistent with both the neg-stain 2D class average with 2 sNS1 dimers on an HDL (Fig S1c) and with the recent study of Flamand & co-workers showing 1-3 NS1 dimers per HDL (ref 20). It is also ignores the propensity of NS1 to associate with membranes and lipids. It is far more likely that NS1 association with HDL is driven by these hydrophobic interactions than by specific protein-protein contacts. A lengthy Discussion section (lines 461-522) includes several chemically dubious or inconsistent statements, all based on the assumption that specific ApoA1 contacts are essential to NS1 association with HDL and that sNS1 oligomers higher than the dimer necessarily involve ApoA1 interaction, conclusions that are not established by the data in this paper.

We thank the Reviewer and have revised our discussion to cover available structural and functional data to draw conclusions that invariably also need further validation by others. One point that is repeatedly brought up by Reviewer 1 & 2 is the quality and functionality of our sample. Our conclusion now reiterates this point based on our own published data (Chan et al., 2019) and also the TEER assay data provided as Author response image 3.

**Reviewer #1 (Recommendations For The Authors):**
Minor:(1) Fig. S3B, should the label for lane 4 be isNS1? In figure 1C you do not see ApoA1 for rsNS1 but for S3B you do? Which is correct?

This has been corrected in the Fig. S3B, the label for lane 4 has been corrected to isNS1 and lane 1 to rsNS1, where no ApoA1 band (25 kDa) is found.

(2) Line 436, is this the correct reference? Reference 43?

This has been corrected in the main text. (p 20, Line 507; Lee et al., 2020, J Exp Med).

**Reviewer #2 (Recommendations For The Authors):**
The cryoEM data analysis is incompletely described. The process (software, etc) leading to each refined EM map should be stated, including the use of reference structures in any step. These details are not in the Methods or in Figs S4-7, as claimed in the Methods. The use of DeepEMhancer (which refinements?) with the lack of defined secondary structural features in the maps and without any validation (or discussion of what was used as "ground truth") is concerning. At the least, the authors should show pre- and post-DeepEMhancer maps in the supplemental figures.

The data processing steps in the Methods section have been described with improved clarity. DeepEMhancer is a deep learning solution for cryo-EM volume post-processing to reduce noise levels and obtain more detailed versions of the experimental maps (Sanchez-Garcia, et al., 2021). DeepEMhancer was only used to sharpen the maps and reduce the noise for classes 1 and 2 of isNS1wt in complex with Ab56.2 for visualization purpose only and not for any refinements. To avoid any confusion, the use of DeepEMhancer has been removed from the supp text and figures.

Line 83 - "cryoEM structures...recently reported" isn't ref 17

This reference has been corrected in to Shu et al. (2022) in p 3, line 83.

Fig. S3 - mis-labeled gel lanes

This has been corrected in the Fig. S3B, the label for lane 4 has been corrected to isNS1 and lane 1 to rsNS1.

Fig S6c caption - "Representative 2D classes of each 3D classes, white bar 100 Å. Refined 3D map for classes 1 and 2 coloured by local resolution". The first sentence is unclear, and there is no white scale bar and no heat map.

Fig S6c caption has been corrected to “Representative 3D classes contoured at 0.06 and its particle distribution as labelled and coloured in cyan. Scale bar of 100 Å as shown. Refined 3D maps and their respective FSC resolution charts and posterior precision directional distribution as generated in crysosparc4.0”.